# The Evolution of Statistical Induction Heads: In-Context Learning Markov Chains

**Ezra Edelman**[*]
University of Pennsylvania
ezrae@cis.upenn.edu

**Nikolaos Tsilivis**[*]
New York University[†]
nt2231@nyu.edu

**Benjamin L. Edelman**
Harvard University
bedelman@g.harvard.edu

**Eran Malach**
Harvard University
emalach@g.harvard.edu

**Surbhi Goel**
University of Pennsylvania
surbhig@cis.upenn.edu

## Abstract

Large language models have the ability to generate text that mimics patterns in their inputs. We introduce a simple Markov Chain sequence modeling task in order to study how this in-context learning capability emerges. In our setting, each example is sampled from a Markov chain drawn from a prior distribution over Markov chains. Transformers trained on this task form *statistical induction heads* which compute accurate next-token probabilities given the bigram statistics of the context. During the course of training, models pass through multiple phases: after an initial stage in which predictions are uniform, they learn to sub-optimally predict using in-context single-token statistics (unigrams); then, there is a rapid phase transition to the correct in-context bigram solution. We conduct an empirical and theoretical investigation of this multi-phase process, showing how successful learning results from the interaction between the transformer's layers, and uncovering evidence that the presence of the simpler unigram solution may delay formation of the final bigram solution. We examine how learning is affected by varying the prior distribution over Markov chains, and consider the generalization of our in-context learning of Markov chains (ICL-MC) task to $n$-grams for $n > 2$.

## 1 Introduction

Large language models (LLMs) exhibit a remarkable ability to perform *in-context learning* (ICL) from patterns in their input context [12, 16]. The ability of LLMs to adaptively learn from context is profoundly useful, yet the underlying mechanisms of this emergent capability are not fully understood.

In an effort to better understand ICL, some recent works propose to study ICL in controlled synthetic settings—in particular, training transformers on mathematically defined tasks which require learning from the input context. For example, a recent line of works studies the ability of transformers to perform ICL of standard supervised learning problems such as linear regression [3, 20, 26, 41]. Studying these well-understood synthetic learning tasks enables fine-grained control over the data distribution, allows for comparisons with established supervised learning algorithms, and facilitates the examination of the in-context "algorithm" implemented by the network. That said, these supervised settings are reflective specifically of *few-shot learning*, which is only a special case of the more general phenomenon of networks incorporating patterns from their context into their

---

[*]Equal Contribution
[†]Work done while visiting Harvard University.

38th Conference on Neural Information Processing Systems (NeurIPS 2024).

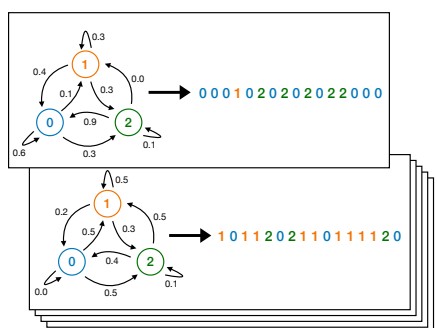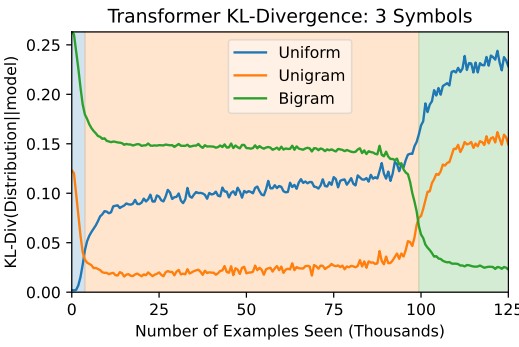

Figure 1: (*left*) We train small transformers to perform in-context learning of Markov chains (ICL-MC). Each training sequence is generated by sampling a transition matrix from a prior distribution, and then sampling a sequence from this Markov chain. (*right*) Distance of a transformer's output distribution to several well-defined strategies over the course of training on our in-context Markov chain task. The model passes through three stages: (1) predicting a uniform distribution (*blue* region), (2) predicting based on in-context unigram statistics (*orange* region), (3) predicting based on in-context bigram statistics (*green* region). Shading is based on the minimum of the curves.

predictions. A few recent works [8, 42] go beyond the case of cleanly separated in-context inputs and outputs, studying in-context learning on distributions based on discrete stochastic processes.

The goal of this work is to propose and analyze a simple synthetic setting for studying ICL. To achieve this, we consider $n$-gram models [11, 15, 37], one of the simplest and oldest methods for language modeling. An $n$-gram language model predicts the probability of a token based on the preceding $n-1$ tokens, using fixed-size chunks ($n$-grams) of text data to capture linguistic patterns. Our work studies ICL of $n$-gram models, where the network needs to compute the conditional probability of the next token based on the statistics of the tokens observed in the input context, rather than on the statistics of the entire training data. We mainly focus on the simple case of $n = 2$; i.e., bigram models, which can be represented as Markov chains. We therefore consider ICL of Markov chains (ICL-MC): we train two layer attention-only transformers on sequences of tokens, where each sequence is produced by a different Markov chain, generated using a different transition matrix (see Figure 1 (left)).

By studying ICL-MC, we are able to replicate and study multiple phenomena that have been observed in ICL for LLMs, and identify new ones. We demonstrate our findings using a combination of empirical observations on transformers trained from scratch on ICL-MC and theoretical analysis of a simplified linear transformer. Our key findings are summarized below:

**(1) Transformers learn statistical induction heads to optimally solve ICL-MC.** Prior work studying ICL in transformers revealed the formation of *induction heads* [18], a circuit that looks for recent occurrence(s) of the current token, and boosts the probabilities of tokens which followed in the input context. We show that in order to solve ICL-MC, transformers learn *statistical* induction heads that are able to compute the correct *conditional (posterior) probability* of the next token given all previous occurrences of the prior token (see attention patterns in Figure 2). We show that these statistical induction heads lead to the transformer achieving performance approaching that of the Bayes-optimal predictor.

**(2) Transformers learn predictors of increasing complexity and undergo a phase transition when increasing complexity.** We observe that transformers display *phase transitions* when learning Markov chains—learning appears to be separated into phases, with fast drops in loss between the phases. We are able to show that different phases correspond to learning models of increased complexity—unigrams, then bigrams (see Figure 1)—and characterize the transition between the phases. We also consider the $n$-gram generalization of our setting, where the next token is generated based on the previous $n-1$ tokens, and observe a similar multi-stage learning process.

**(3) Simplicity bias may slow down learning.** We provide evidence that the model's inherent bias towards simpler solutions (in particular, in-context unigrams) causes learning of the optimal solution

to be delayed. Changing the distribution of the in-context examples to remove the usefulness of in-context unigrams leads to faster convergence, even when evaluated on the original distribution.

**(4) Alignment of layers is crucial.** We show that the transition from a phase of learning the simple-but-inadequate solution to the complex-and-correct solution happens due to an alignment between the layers of the model: the learning signal for the first layer is tied to the extent to which the second layer approaches its correct weights.

## 1.1 Related Work

**In-Context Learning.** In [13], the authors discuss how properties of the data distribution promote ICL. Xie et al. [42] suggest a Bayesian interpretation of ICL and studies how ICL emerges when the training distribution comes from a Hidden Markov Model (HMM). Abernethy et al. [2] study the ability of transformers to segment the context into pairs of examples and labels and provide learning guarantees when the labeling is of the form of a sparse function. The work of Bietti et al. [8] studies the dynamics of training transformers on a task that is reminiscent of our Markov chain setting but has additional complexities. Instead of drawing a fresh Markov chain for each sequence, in their task all sequences are sampled from the same Markov chain; after certain 'trigger' tokens, the following 'output' token is chosen deterministically within a sequence. Thus, successful prediction requires incorporating both global bigram statistics and in-context deterministic bigram copying, unlike in our setting where the patterns computed by *statistical* induction heads are necessary and sufficient. As in our work, the authors identify multiple distinct stages of training and show how multiple top-down gradient steps lead to a solution.

**Induction Heads.** Elhage et al. [18] relates ICL with the formation of induction heads, sub-components of transformers that match previous occurrences of the current token, retrieving the token that succeeds the most recent occurrence. Reddy [34] studies the formation of induction heads and their role in ICL, showing empirically that a three layer network exhibits a sudden formation of induction heads towards solving some ICL problem of interest. Bietti et al. [8] study the effect of specific trigger tokens on the formation of induction heads.

**Phase Transitions.** It has been observed in different contexts that neural networks and language models display a sudden drop in loss during their training process. This phase transition is often related to emergence of new capabilities in the network. The work of Power et al. [32] observed the "grokking" phenomenon, where the test loss of neural networks sharply drops, long after the network overfits the training data. Chen et al. [14] shows another example of a phase transition in language model training, where the formation of specific attention mechanisms happen suddenly in training, causing the loss to quickly drop. Barak et al. [7] observe that neural networks trained on complex learning problems display a phase transition when converging to the correct solution. Several works [25, 27] attribute these phase transitions to rapid changes in the inductive bias of networks, while Merrill et al. [29] argue that the models are sparser after the phase change. Schaeffer et al. [35] warn that phenomena in deep learning that seem to be discontinuous can actually be understood to evolve continuously once seen through the right lens.

**Simplicity Bias.** Various works observed that neural networks have a "simplicity bias", which causes them to "prioritize" learning simple patterns first [5, 39]. The work of Kalimeris et al. [23] shows that SGD learns functions of increased complexity, first fitting a linear concept to the data before moving to more complex functions. [36] shows that the simplicity bias of neural networks can sometimes be harmful, causing them to ignore important features of the data. Chen et al. [14] demonstrate the effect of simplicity bias on language tasks that require understanding of syntactic structure. Abbe et al. [1] provide a theoretical framework for understanding how the simplicity of the target function can govern the convergence time of SGD, describing how simple partial solutions can speed up learning; in contrast, in our setting, the unigram solution appears likely to be a distractor which delays learning of the correct solution.

**Concurrent works.** In parallel to this work, there have been a number of papers devoted to the study of similar questions regarding in-context learning or Markov chains: Akyürek et al. [4] empirically compare the ability of different architectures to perform in-context learning of regular languages. Their experiments with synthetic languages motivate architectural changes which improve natural language modeling in large scale datasets. Hoogland et al. [21] observe similar stage-wise learning behaviors on transformers trained on language or synthetic linear regression tasks. Makkuva et al.

[28] study the loss landscape of transformers trained on sequences sampled from a single Markov Chain. Perhaps closest to our work, Nichani et al. [30] introduces a general family of in-context learning tasks with causal structure, a special case of which is in-context Markov chains. The authors prove that a simplified transformer architecture (similar to the one we introduce in Section 2.2) can learn to identify the causal relationships by training via gradient descent, and also characterize the ability of the trained models to adapt to out-of-distribution data. The focus of our work, instead, is on the different stages of training and how they relate to specific, well-defined, strategies.

## 2  Setup

In this section, we describe our learning problem and present the neural network architectures that we will use for learning.

**ICL-MC Task.** Our learning task consists of sequences generated from Markov Chains with random transition matrices. The goal is to in-context estimate the transition probabilities from sampled sequences, in order to predict the next state. Formally, each sample sequence is generated by a Markov Chain with state space $S = \{1, \ldots, k\}$ and a transition matrix $\mathcal{P}$ sampled from a prior distribution, with $x_1$ drawn from some other prior distribution (potentially dependent on $\mathcal{P}$), and the rest of $\boldsymbol{x} = (x_1, \ldots, x_T)$ drawn from the Markov Chain. We primarily focus on the case where each row of the matrix is sampled from the Dirichlet distribution with concentration parameter $\boldsymbol{\alpha}$, i.e. $\mathcal{P}_{i,:} \sim \text{Dir}(\boldsymbol{\alpha})$. We want to learn a predictor that, given context $x_1, \ldots, x_T$, predicts the next token, $x_{T+1}$. Note that this is an inherently non-deterministic task, even provided full information about the transition matrix, and as such it can better capture certain properties of language than previous in-context learning modeling approaches, such as linear regression [20]. We focus on the case of the *flat* Dirichlet distribution, with $\boldsymbol{\alpha} = (1, \ldots, 1)^\top$, that corresponds to uniform transition probabilities between states. We draw the initial state $x_1$ from the stationary distribution $\boldsymbol{\pi}$ of the chain (which exists almost surely). We primarily consider the case where the number of states $k$ is 2 or 3.

In subsection 3.3, we consider the generalization of this setting to $n$-grams for $n > 2$. Instead of the distribution of $x_T$ being determined by $x_{T-1}$, we let it be determined by $x_{T-n+1}, \ldots, x_{T-1}$, according to a conditional distribution $\mathcal{P}$ which is uniform over the possible states[3].

### 2.1  Potential Strategies for (Partially) Solving ICL-MC

We adopt the Bayesian interpretation of in-context learning [42], in which a prior distribution is provided by the training data, and, at test time, the model updates this prior given the in-context sequence. In this framework, we focus on two strategies for Bayesian inference: a (suboptimal) *unigram* strategy which assumes tokens in each sequence are i.i.d. samples (and counts the frequency of the states in the sequence so far), and the *bigram* strategy which correctly takes into account dependencies among adjacent tokens (and counts frequency of pairs of tokens).

**1st strategy: Unigrams.** Since we initialize the Markov chain at its stationary distribution (which exists a.s.), the optimal strategy across unigrams is just to count frequency of states and form a posterior belief about the stationary distribution. Unfortunately, the stationary distribution of this random Markov chain does not admit a simple analytical characterization when there is a finite number of states, but it can be estimated approximately. At the limit of $k \to \infty$, the stationary distribution converges to the uniform distribution [10].

**2nd strategy: Bigrams.** For any pair of states $i$ and $j$, let $\mathcal{P}_{ij}$ be the probability of transitioning from $i$ to $j$. On each sample $\boldsymbol{x}$, we can focus on the transitions from the $i$-th state, which follow a categorical distribution with probabilities equal to $(\mathcal{P}_{i1}, \ldots, \mathcal{P}_{ik})$. If we observe the in-context empirical counts $\{c_{ij}\}_{j=1}^k$ of the transitions, then $\mathcal{P}_{ij}$ is given by: $(\mathcal{P}_{i1}, \ldots, \mathcal{P}_{ik}) | \boldsymbol{x} \sim \text{Dir}(k, c_{i1} + \alpha_1, \ldots, c_{ik} + \alpha_k)$, where $\alpha_1, \ldots, \alpha_k$ are the Dirichlet concentration parameters of the prior. Hence, each $\mathcal{P}_{ij}$ has a (marginal) distribution that is actually a Beta distribution: $\mathcal{P}_{ij} | \boldsymbol{x} \sim \text{Beta}\left(c_{ij} + \alpha_j, \sum_j \alpha_j + N_i - \alpha_j - c_{ij}\right)$, where $N_i$ is the total number of observed transi-

---

[3]In particular, for each tuple of $n - 1$ tokens, we sample the vector of conditional probabilities for the next state from a flat Dirichlet distribution.

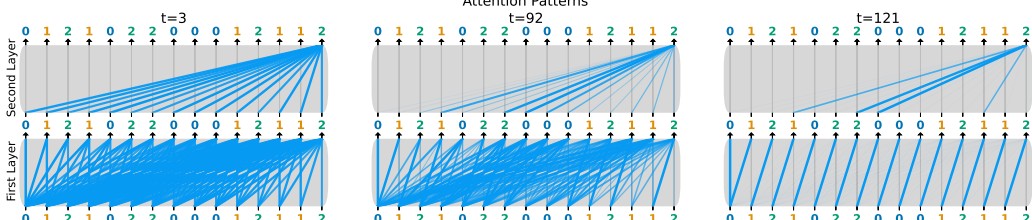

Figure 2: Attention patterns that correspond to the last token of the sequence for a transformer trained to perform ICL-MC. The intensity of each blue line signifies the strength of the corresponding attention value. As the model gets trained, we observe that the attention weights mimic the construction of Proposition 2.1. Specifically, at the end of training (*right*), each token in the first layer is attending to the previous token. In the second layer, the last token, a "2", is attending to tokens that followed "2"s, allowing bigram statistics to be calculated. See also Figure 9 for full attention matrices during the course of training.

tions from state $i$. As such, our best (point) estimate for each state $j$ is given by: $\mathbb{E}\left[\mathcal{P}_{ij}|\boldsymbol{x}\right] = \frac{c_{ij}+\alpha_j}{N+\sum_i \alpha_i}$. For the uniform Dirichlet, $\boldsymbol{\alpha} = (1, \dots, 1)^\top$, it is $\mathbb{E}\left[\mathcal{P}_{ij}|\boldsymbol{x}\right] = \frac{c_{ij}+1}{N_i+k}$.

## 2.2 Architectures: Transformers and Simplifications

We are interested in investigating how transformers [40] can succeed in in-context learning this task. We focus on attention-only transformers with 2 layers with causal masking which is a popular architecture for language modeling. Given an input sequence $\boldsymbol{x}$, the output of an $n$-layer attention-only transformer[4] is:

$$TF(E) = P \circ (Attn_n + I) \cdots \circ (Attn_1 + I) \circ E, \tag{1}$$

where $E \in \mathbb{R}^{T \times d}$ is an embedding of $\boldsymbol{x} \in \mathbb{R}^d$, $P \in \mathbb{R}^{d \times k}$ is a linear projection to the output logits, and $Attn(\boldsymbol{x})$ is masked self attention with relative position embeddings [38], which is parameterized by $W_Q, W_K, W_V \in \mathbb{R}^{d \times d}, v \in \mathbb{R}^{T \times d}$:

$$Attn(z) = \text{softmax}(\text{mask}(A))zW_V, \qquad A_{i,j} = \frac{(z_i W_Q)(z_j W_K + v_{i-j+1})^\top}{\sqrt{d}}. \tag{2}$$

In general, transformers often contain an MLP module, but for this task they are not necessary (see Appendix A and Figure 10 for additional experiments with transformers with MLPs). During training, we minimize the loss:

$$L(\theta) = \mathop{\mathbb{E}}_{\substack{\boldsymbol{x} \sim \mathcal{P} \\ \mathcal{P} \sim \text{Dir}(\boldsymbol{\alpha})^k}} \left[ \frac{1}{t} \sum_{i=1}^{T} l\left(TF(\boldsymbol{x}; \theta)_i, x_{i+1}\right) \right], \tag{3}$$

where $\theta$ denotes the parameters of the model and $l$ is the cross entropy loss. Notice that we provide supervision in all positions, as standard in language modeling.

We now show how a two-layer transformer of the above architecture can represent the optimal bigrams solution.

**Proposition 2.1** (Transformer Construction). *A single-head two layer attention-only transformer can find the bigram statistics in the in-context learning Markov chain task.*

Intuitively, the first layer of the transformer copies the previous token at each position, and in the second layer each token sums the embeddings of all the tokens whose output from the first layer matches itself. The full proof can be found in Appendix B.1.

**Simplified Transformer Architecture.** As we see from the construction, there are two main ingredients in the solution realized by the transformer; (1st layer) the ability to look one token back and (2nd layer) the ability to attend to itself. For this reason, we define a *minimal model* that is

---

[4]For simplicity, we assume embedding and hidden dimension are equal, but they can be different in general.

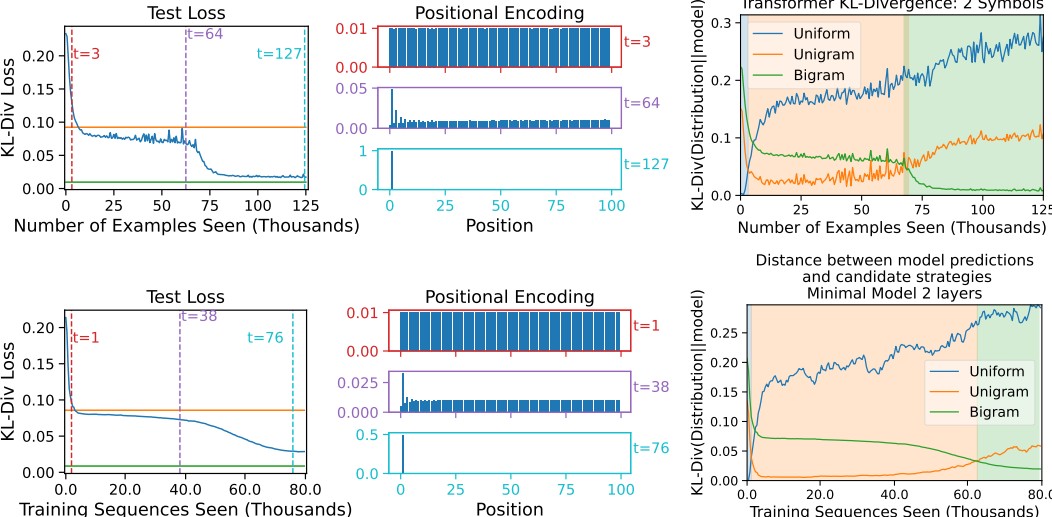

Figure 3: A two layer transformer (*top*) and a minimal model (*bottom*) trained on our in-context Markov Chain task. A comparison of the two layer attention-only transformer and minimal model (4) (with $v$ and $W$ initialized to 0). The graphs on the left are test loss measured by KL-Divergence from the underlying truth. The orange line shows the loss of the unigram strategy, and the green line shows the loss of the bigram strategy. The middle graph shows the effective positional encoding (for the transformer, these are for the first layer, and averaged over all tokens). The graph on the right shows the KL-divergence between the outputs of the models and three strategy. The lower the KL-divergence, the more similar the model is to that strategy.

expressive enough to be able to represent such a solution, but also simple enough to be amenable to analysis. Let $e_{x_i}$ denote the one-hot embedding that corresponds to the state at position $i \in [T]$, and let $E$ be the $\mathbb{R}^{(T+1) \times k}$ one-hot embedding matrix. Then the model is parameterized by $W \in \mathbb{R}^{k \times k}$ and $v \in \mathbb{R}^{T+1}$ and defined as:

$$f(E) = \text{mask}(EW(\text{Softmax}(M)E)^{\top})E, \quad M = \begin{pmatrix} v_0 & -\infty & \dots & -\infty \\ v_1 & v_0 & \dots & -\infty \\ \vdots & \vdots & \dots & \vdots \\ v_T & v_{T-1} & \dots & v_0 \end{pmatrix} \in \mathbb{R}^{(T+1) \times (T+1)},$$

(4)

where $\text{mask}(\cdot)$ is a causal mask, and $\text{Softmax}(M)_{i,j} = \frac{\exp(M_{i,j})}{\sum_{t=0}^{T} \exp(M_{T,j})}$. Notice that the role of $W$ is to mimic the attention mechanism of the second layer and the role of $v$ is that of the relative positional embeddings. This model can be seen as a simplified version of a two-layer linear attention-only transformer. See also Appendix B.2 for a discussion.

*Fact* 2.2. Both the bigrams strategy and the unigrams strategy can be expressed by the minimal model with a simple choice of weights.

- **Bigrams:** Let $v = \begin{pmatrix} 0 & c & 0 & \dots & 0 \end{pmatrix}^{\top}$ and $W = I_{k \times k}$, then $f(E)_{T,s} = \sum_{t=2}^{T} \mathbb{1}\{x_t = s\} \mathbb{1}\{x_{t-1} = x_T\} + O\left(\frac{kT^2}{\exp(c)}\right)$.

- **Unigrams:** For $v = \begin{pmatrix} 0 & \dots & 0 \end{pmatrix}^{\top}$, $W = \frac{1}{k} \mathbb{1}\mathbb{1}^{\top}$, we have $f(E)_{T,s} = \sum_{t=1}^{T} \mathbb{1}\{x_t = s\}$.

See Section B for the proofs.

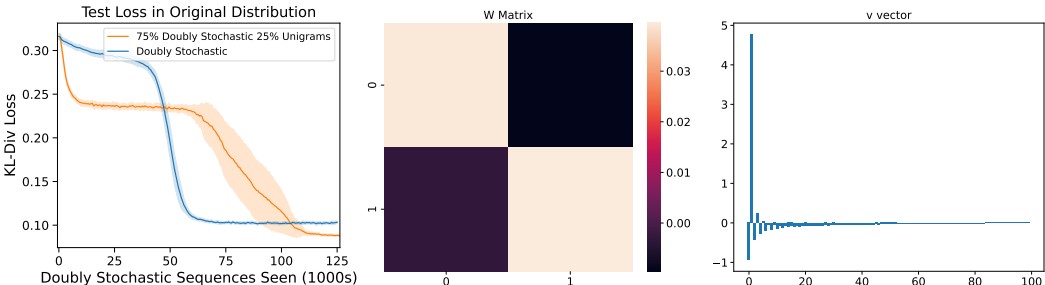

Figure 4: (*left*) Unigrams slow down optimization: Comparison of two-layer attention only transformers trained on two distributions; one with a uniformly random doubly stochastic transition matrix and another with a mixture of the doubly stochastic and unigrams distribution (see Appendix A.1 for details). We see that in absence of unigrams "signal" the model minimizes the loss (evaluated on the full distribution) much faster. (*center, right*) Training of the minimal model on ICL-MC with $k = 2$ states: (*center*) The heatmap of the second layer ($W$ matrix) that learns to be close to diagonal. (*right*) The values of the positional embeddings (1st layer) that display a curious even/odd pattern. This is before any softmax is applied to the positional embeddings.

## 3 Empirical Findings and Theoretical Validation

In this section, we present our empirical findings on how transformers succeed in in-context learning Markov Chains[5], we demonstrate the different learning stages during training and the sudden transitions between them, and draw analytical and empirical insights from the minimal model.

### 3.1 Transformers In-Context Learn Markov Chains Hierarchically

We focus on attention-only transformers with 2 layers with causal masking and relative positional encodings and train them with the Adam optimizer on ICL-MC (see Section A for experimental details). As can be seen in Figure 3, all the models converge near the Bayesian optimal solution, suggesting that they learn to implement the bigram strategy. Curiously, however, learning seems to be happening in stages; there is an initial rapid drop and the model quickly finds a better than random solution. Afterwards, there is a long period of only slight improvement before a second rapid drop brings the model close to the Bayes optimal loss. We observe that training a 1-layer transformer fails to undergo a phase transition or converge to the right solution, even if trained for 10x the amount of time - see Figure 7 in the Appendix.

Interestingly, as can be seen from the horizontal lines in Figure 3, the intermediate plateau corresponds to a phase when the model reaches the unigram baseline. We provide evidence that this is not a coincidence, and that after the initial drop in loss, the model's strategy is very similar to the unigram strategy, before eventually being overtaken by the bigram strategy. Some of the strongest such evidence is on the right in Figure 3, where we plot the KL divergence between model's prediction and the two different strategies. For both the strategies, their KL divergence from the model quickly goes down, with the unigram solution being significantly lower. Around the point of the second loss drop, the KL divergence between the model and the bigram solution decreases, while the other one increases, making it clear that the model transitions from the one solution to the other. This final drop is what has been associated to prior work with *induction heads* formation [31]; special dedicated heads inside a transformer are suddenly being formed to facilitate in-context learning. Similar observation hold for Markov Chains with a larger number of states - see Figures 8 and 11.

**Mechanistic evidence for solutions found by transformer.** To confirm how the two layer attention-only transformer solves ICL-MC, we inspected the attention in each layer throughout training. Figure 2 shows the attention for a particular input during different parts of training. By the end of training, the attention patterns match that of our construction in Proposition 2.1, with the first layer attending to tokens one in the past, and the second layer attending to tokens that follow the same token as the current one.

---

[5]Code: `https://github.com/EzraEdelman/Evolution-of-Statistical-Induction-Heads`.

**Varying the data distribution - Unigrams slow down learning.** There are several interesting phenomena in the learning scenario that we just described, but it is the second drop (and the preceding plateau) that warrants the most investigation. In particular, one can ask the question: *is the unigram solution helpful for the eventual convergence of the model, or is it perhaps just a by-product of the learning procedure?* To answer these questions, we define distributions over Markov chains that are in between the distribution where unigrams is Bayes optimal, and the distribution where unigrams is as good as uniform - see Appendix A for more details. As we see in Figure 4, the transformers that are being trained on the distribution where there is no unigrams "signal" train much faster. And even more tellingly, giving additional "unigram samples" curiously slows down learning. See also Figure 12 in the Appendix that displays how the models perform on different parts of the distribution during training.

## 3.2 Theoretical Insights from the Minimal Model

To abstract away some of the many complicated components from the transformer architecture, we focus our attention now to the minimal model of Section 2.2. We train minimal models of eq. (4), starting from a deterministic constant initialization, by minimizing the cross entropy loss with SGD. Full experimental details can be found in Appendix A. Figure 3 (bottom) displays the training curves for the minimal model. Similar to the transformer, learning occurs in two stages and the models eventually converge close to the optimal solution.

We now provide theoretical insights on how training progresses stage by stage and how this is achieved by the synergy between the two layers. As it turns out, there need to be at least two steps of gradient descent in order for both elements of the solution to be formed.

**Proposition 3.1.** *Let the model be defined as in eq.* (4) *and initialized with* $W^{(0)} = \mathbf{0}, v^{(0)} = \mathbf{0}$. *Suppose the transition matrix* $P \in \mathbb{R}^{k \times k}$ *is sampled from one of the following two types of distribution:*

1. *$k = 2$, and $P$ is sampled from the uniform distribution over the set of $2 \times 2$ stochastic matrices.*
2. *For any constant $k$ and $0 < \alpha < 1$, with probability $\alpha$, sample the matrix $P$ uniformly from a "bigram-only" distribution—the set of $k \times k$ doubly stochastic matrices; and with probability $1 - \alpha$ use a "unigram-only distribution": draw a vector $u$ uniformly from the set $\{u \in \mathbb{R}_{\geq 0}^k : \|u\|_1 = 1\}$ and let $P = \mathbf{1}u^\top$.*

*Then after one step of population gradient descent with step size $\eta > 0$,*

$$W^{(1)} = \Theta(\eta T)I + \Theta(\eta T)\mathbf{1}\mathbf{1}^\top + E \text{ and } v^{(1)} = \mathbf{0}$$

*where $\|E\| \leq O(\eta \log T)$. In other words, for large $T$, the second layer weights are a mixture of the correct solution $I$ and uniform attention $\mathbf{1}\mathbf{1}^\top$.*

*Assuming in the first step $\eta_1 = O\left(\frac{1}{T^2}\right)$, then $W^{(2)}$ has the same structure as $W^{(1)}$ (up to scaling). Furthermore,*

$$v_1^{(2)} = \Omega(\eta_2 \log T), \text{ and } v_n^{(2)} \leq cv_1^{(2)} \ \forall n \neq 1, c < 1$$

*where $\eta_2$ is the step size for the second step.*

*If $\eta_2 = \Omega(T)$, then the output of the model will be a weighted sum of bigrams and unigrams strategy. Formally,*

$$f(E)_{T,s} = \Theta(\eta_2 T)\sum_{i=1}^{T} \mathbb{1}\left[x_{i-1} = x_t, x_i = s\right] + \Theta(\eta_2 T)\sum_{i=1}^{T} \mathbb{1}\left[x_i = s\right] + O(\log T)$$

**Note** In the first distribution (uniformly random $2 \times 2$) or the second distribution with $k > 6$, at the end of the two steps, the weight on bigrams is greater than that of the weight on unigrams strategy.

*Proof Overview.* The idea of the proof is that a first step of gradient descent with a small learning rate can align the second layer, while a second step can learn to identify the correct relative positional embedding. The identity bias of $W$ in the second layer ensures there is a strong signal in the gradient to look back one in the first layer. Without a bias in $W$, the gradient for the positional encodings, $v$, is zero.

We get additional intuition from the second distribution (mixture of unigrams and doubly stochastic): in the first step, effectively all of the gradient comes from the examples where the unigram strategy is optimal, while in the second step effectively all of the gradient comes from the examples where the bigram strategy is optimal.

*Remark* 3.2. It is worth noting that, while this is a simplified setting, the analysis goes beyond NTK-based [22] analyses where the representations do not change much and it crucially involves more than one step which has been a standard tool in the analysis of feature learning [6].

We summarize the key theoretical implications:

**Learning occurs in two phases.** Both in the theoretical and experimental models, training has two phases that work at very different speeds. The first phase is fast in both cases; in the theoretical setting, even a single step with step size $O\left(\frac{1}{T}\right)$ is sufficient for learning the second layer. In the second phase, a much larger step size of $\Omega(1)$ is needed in order to learn the positional encodings (in one step).

**Second layer is learned first.** It has been observed before in a similar bigram learning setting with a two-layer transformer that the model might be learning first the second layer [8]. We also make similar observations in our experiments with the minimal model and the transformers (see Figure 2). For the minimal model, the gradient calculations, clearly suggest that starting from a default initialization, it is only the second layer that quickly "picks up" the right solution.

**Even/odd pattern in positional encodings emerges.** We notice in the experiments, that the positional embeddings of both the transformer and minimal model displayed an intriguing even/odd oscillating pattern - see Figure 3 (*top, center*) and Figure 4 (*right*). We believe that a careful analysis the gradient of $v$ in the second step will recover this pattern, which is likely related to the moments of the eigenvalues of the transition matrix.

### 3.3 Beyond Bigrams: $n$-gram Statistics

Finally, we investigate the performance of transformers on learning in-context $n$-grams for $n > 2$; in particular, 3-grams. We train attention-only transformers with three heads in each layer[6] by minimizing the in-context cross entropy loss with the Adam optimizer. As can be seen in Figure 5 (left), the model eventually converges to the Bayes optimal solution. Interestingly, as in the case of Markov Chains, the model displays a "hierarchical learning" behavior characterized by long plateaus and sudden drops. In this setup, the different strategies correspond to unigrams, bigrams and trigrams, respectively. This is presented clearly on the right of Figure 5, where we plot the similarity of the model with the different strategies and it exhibits the same clear pattern as in the case of $n = 2$. We leave a more thorough investigation of $n$-grams for future work.

## 4 Conclusion and Discussion

In this work, we have introduced a simple learning problem which serves as a controlled setting for understanding in-context learning and the emergence of (statistical) induction heads. Through a combination of theoretical analysis and empirical investigation, we identify different stages during learning, which we were able to precisely characterize. These validate similar observations from training large-scale language models.

The main limitation of our work is that our analysis relies on a simplified transformer architecture and our learning task is synthetic. Yet, we see the simplicity of our modeling as a positive, since it allows to make rigorous predictions about the mechanisms behind in-context learning abilities of a transformer. On the theoretical front, it would be interesting to extend our analysis to handle higher number of symbols and more complex models for language generation (beyond Markov chains).

On the empirical front, it would be worthwhile to understand similar stage-wise learning with natural language data, and use insights from our minimal model to improve formation of induction

---

[6]Our experiments with one head per layer did not converge to low loss during training, but follow-up work by [33] showed that with sufficiently long training time a single head in each layer can suffice.

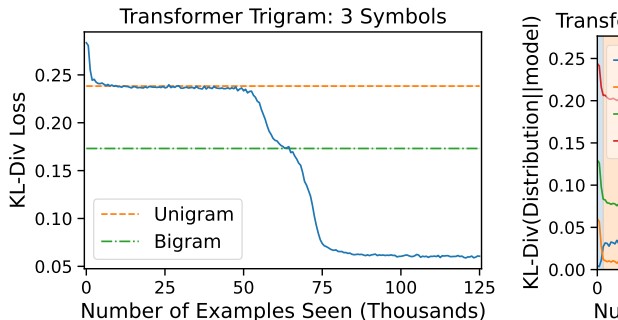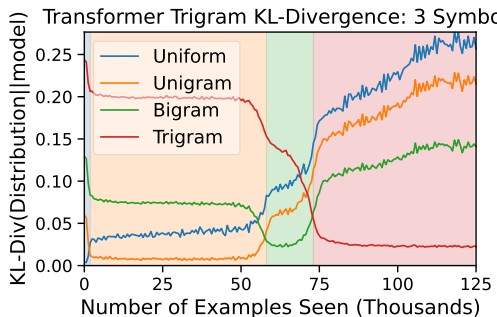

Figure 5: Three-headed transformer trained on In-Context Learning 3-grams (trigrams), with context length 200. (*left*) Loss during training. The model hierarchically converges close to the Bayes optimal solution. (*right*) KL divergence between the model and different strategies during training. As we observe, there are 4 stages of learning, each of them corresponding to a different algorithm implemented by the model.

heads. In particular, it would be great to understand if better data curriculum could remove the undesirable simplicity bias we observe from unigrams. Such simple but incomplete solutions may be commonplace in language modeling and other rich learning settings; for any such solution, one can ask to what extent its presence speeds up or slows down the formation of more complex circuits with higher accuracy.

**Acknowledgements.** EE thanks Alan Yan for helpful conversations. NT acknowledges support through the National Science Foundation under NSF Award 1922658. NT would like to thank Boaz Barak, Cengiz Pehlevan and the whole ML Foundations Group at Harvard for their hospitality during Fall 2023 when most of this work was done. BE acknowledges funding from the ONR under award N00014-22-1-2377 and the NSF under award IIS 2229881. SG acknowledges support through the Open AI SuperAlignment Fast Grants.

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

# A Experimental Details and Additional Experiments

## A.1 Experimental details

We train transformers of the form (1) with the AdamW optimizer with learning rate $3 \times 10^{-5}$ (for 3-grams a learning rate of $3 \times 10^{-2}$ was used), batch size $64$, and hidden dimension $16$. The sequence length of the examples is $100$ tokens. The minimal model was trained with SGD, with batch size $64$, and learning rate $3 \times 10^{-4}$. We use PyTorch 2.1.2.

The data was generated in an online fashion, using numpy.random.dirichlet to generate each row of the transition matrices. At each epoch, we generate a new transition matrix, and, then, a sequence from the induced Markov chain (starting from the stationary distribution of the chain). Both the model initialization (for the transformers) and the data were randomized based on the seed (in a perfectly reproducible manner).

Some of the training and model code was based on minGPT [24]. The experiments all measure the outputs of the models at the last token.

All of the experiments were performed with a single NVIDIA GeForce GTX 1650 Ti GPU with 4 gigabytes of vram with 32 gigabytes of system memory. Each training run took under ten minutes.

The code used can be found at:
`https://github.com/EzraEdelman/Evolution-of-Statistical-Induction-Heads`.

**Details for experiment on the left in Figure 4**    In this experiment, we compared the test loss of the model trained in too different ways. Consider two distributions, each uniform over their support (subsets of the space of $3 \times 3$ transition matrices for this particular experiment):

1. The "doubly stochastic" distribution is uniform over the space of doubly stochastic matrices, that is, transition matrices for which each row or column has entries summing to $1$. This is equivalent to the space of Markov Chains with a uniform stationary distribution (that is, $\pi = \frac{1}{k}\mathbb{1}$), which means that the unigram and uniform strategy are the same.
2. The "unigram distribution" is uniform over the space of stochastic matrices $P$ such that for any two rows $P_i, P_j$ in $P$, $P_i = P_j$. This is equivalent to the distribution over markov chains for which the distribution of the next state doesn't change depending on the previous state. Notice that the unigram strategy is asymptotically optimal on any markov chain from this distribution.

We first train models on a number of samples from the doubly stochastic distribution, and plot their training loss as the blue line. Then, we trained models on a random mixture of the two distributions, with $75\%$ of the samples coming from doubly stochastic distribution, and the remaining $25\%$ coming from the unigram distribution. Importantly, we train this second batch of models on more total samples, so that they see the same number of samples from the doubly stochastic distribution, and then plot their test loss in orange with the x-axis being the number of samples from the doubly stochastic distribution seen. This means that every point on the x-axis in the graph, the models in the orange line have seen more samples than seen by those in the blue line, and yet still take longer to converge.

We would also like to note that the test loss measured is in the distribution uniform over all stochastic matrices, which the models in the orange line do seem to generalize slightly better to after convergence.

**Note on KL-divergence**    In our experiments, we used KL divergence to measure the difference between the probabilities predicted by the model and various other probability distributions. For test loss (depicted, for instance, in Figure 3 (left)), this other distribution comes from the appropriate rows of the transition matrices used to generate the test examples.

Formally, let $f(\boldsymbol{x}_{1:T-1})$ be the softmax distribution of the transformer's output, given the input sequence $\boldsymbol{x}_{1:T-1}$. In our standard setting, we measured

$$d_{KL}(\mathcal{P}_{\boldsymbol{x}_{T-1}} || f(\boldsymbol{x}_{1:T-1}))$$

where $\mathcal{P}_{\boldsymbol{x}_{T-1}}$ is the true distribution of the next state $\boldsymbol{x}_T$ given the previous state, under the true Markov chain $\mathcal{P}$. Note that $\mathcal{P}$ varies from sequence to sequence (it is drawn from a prior over

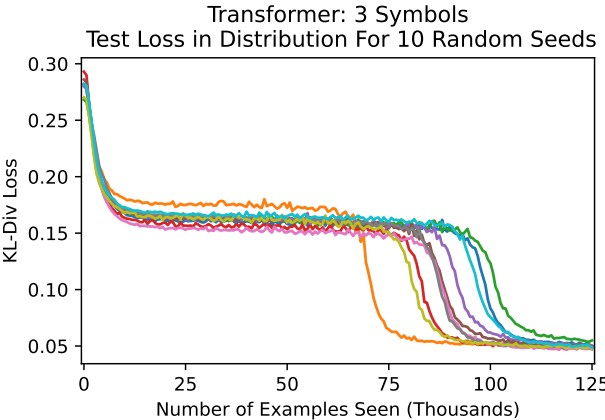

Figure 6: In distribution test loss for 10 two layer attention only transformers, with random seeds $0, 1, \ldots 9$ (randomness affects initialization and the training data). The training dynamics are consistent for each model, though the exact position of the phase transitions changes.

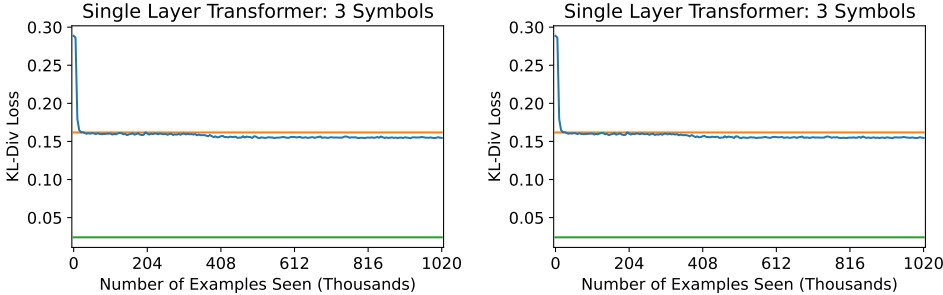

Figure 7: Graphs of test loss showing that a single layer transformer can not achieve good performance on ICL-MC. This result holds for transformers with or without MLPs, and with absolute or relative positional encodings. These graphs show that even trained 8 times longer, there is no notable increase in performance beyond the unigrams strategy (orange line).

transition matrices) and is not directly observable by the learner—this is what needs to be learned in-context.

For measuring how close the model was to various strategies (for instance, in Figure 3 (right)), we computed the predicted probabilities given by said strategies, and used those as the base distribution. Note that the output of the bigrams strategy (which is Bayes-optimal for our base setting) is different from the aforementioned ground-truth $\mathcal{P}_{\boldsymbol{x}_{T-1}}$). Instead, as described in Section 2, it is a Bayesian posterior distribution of the next state given the observed sequence, with the prior determined by the prior distribution of transition matrices. Formally, this corresponds to $\mathbb{E}[\mathcal{P}_{\boldsymbol{x}_{T-1}} | \boldsymbol{x}_{1:T-1}]$, where the expectation is taken over the draw of Markov chain transition matrix.

### A.2 Additional experiments

In the main text, we mainly show experiments with one seed per experiment, in order to keep presentation simple. Figure 6 justifies this choice: it plots the test loss of two-layer transformers (multiple random seeds) trained on ICL-MC for chains with 3 states. Randomness slightly affects the duration of the plateau, but not the qualitative, two-stage, process of learning.

Figure 7 demonstrates the inability of one layer attention only transformers to in-context learn Markov Chains with 3 states.

Figure 8 shows that our results extend for Markov Chains with larger state spaces (here, $k = 8$). As the number of states grows, larger sequence lengths are needed for learning (this is to be expected as a larger transition matrix needs to be estimated in context - roughly, sequence length needs to be $\Omega(k^2)$).

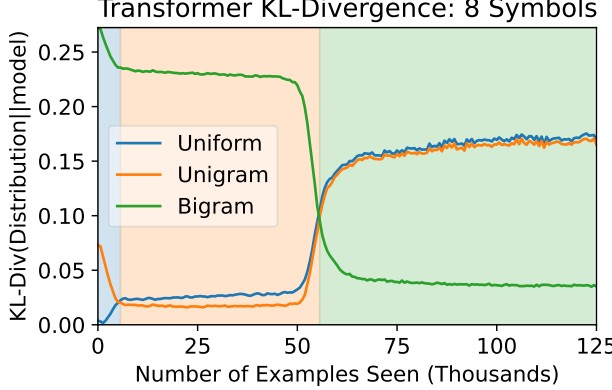

Figure 8: Our results extend to more symbols than $k = 2$ or $k = 3$. The KL-divergence between the transformer and strategies over training. This required a sequence length greater than 100 (200 in this case) for the difference between unigrams and uniform to be large enough for the unigram phase to be visible (in either case there was a plateau before the final drop in test loss).

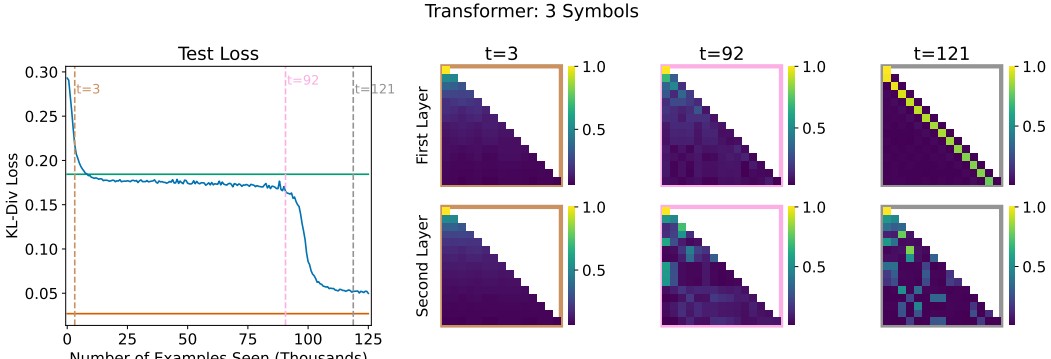

Figure 9: A two layer attention-only transformer trained with cross entropy loss on ICL-MC. The heatmaps on the right represent part of the attention for the transformer at various time steps, specifically the values of the matrix $A$ from (2). The top row are showing $A$ from the first layer, and the bottom row from the second layer.

We also know that, in this particular family of Markov Chains, the stationary distribution approaches the uniform distribution as the number of states grows. As a result, we expect the difference between uniform and unigram solutions to be less noticeable.

Figure 9 shows the attention patterns for the two layers of a transformer during training.

In Figure 10, we demonstrate that our observations extend to two-layer transformers, which have an additional fully-connected MLP on top of the attention layers.

Finally, in Figure 12 we plot the performance of the transformer and the minimal model in various distributions over the course of training.

# B  Proofs

In this section, we present our theoretical results on in-context learning Markov Chains of Section 2.2.

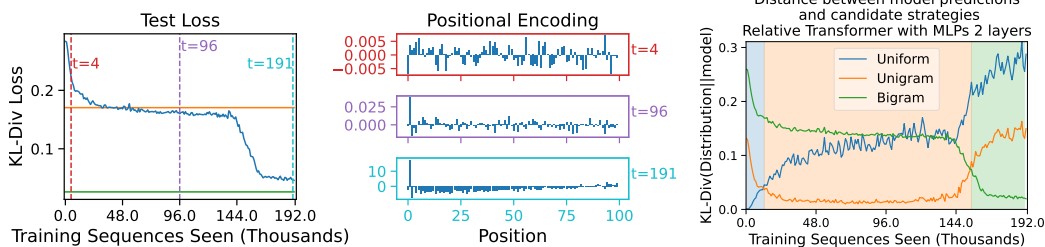

Figure 10: A two layer relative position encoding transformer with MLPs trained on ICL-MC with k=3 symbols. Notice while slightly noisier, the overall trend and observations made regarding the attention only transformer still hold.

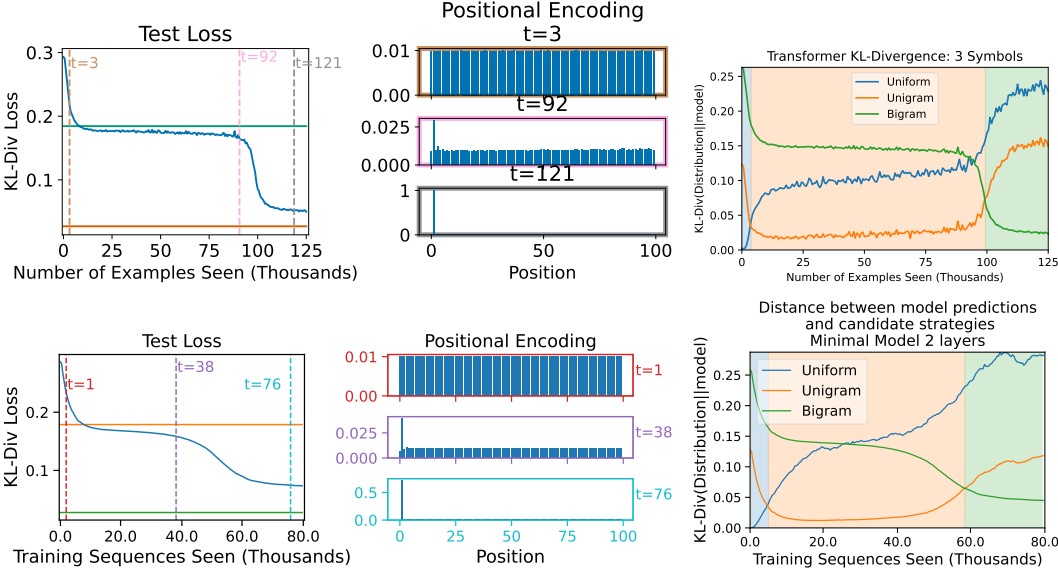

Figure 11: A comparison of the two layer attention only transformer and minimal model for $k = 3$ symbols.

## B.1 Transformer Construction

*Proof of Proposition 2.1.* Set the internal dimension $d = 3k$, and choose $\mathbf{e_x}$ to be one-hot embeddings—that is, $\mathbf{e}_{\boldsymbol{x}_i} = \delta_{\boldsymbol{x}_i}$, where $\delta$ is the Kronecker delta. We will call the parameters of attention layer $i$, $W_Q^{(i)}, W_K^{(i)}, W_V^{(i)}, v^{(i)}$. Let

$$v^{(1)} = \begin{pmatrix} \delta_2 \mathbf{1}_k^\top \\ \mathbf{0} \\ \mathbf{0} \end{pmatrix} \quad W_Q^{(1)} = \begin{pmatrix} cI^{k\times k} & \mathbf{0} & \mathbf{0} \\ \mathbf{0} & \mathbf{0} & \mathbf{0} \\ \mathbf{0} & \mathbf{0} & \mathbf{0} \end{pmatrix} \quad W_K^{(1)} = \mathbf{0} \quad W_V^{(1)} = \begin{pmatrix} \mathbf{0} & I^{k\times k} & \mathbf{0} \\ \mathbf{0} & \mathbf{0} & \mathbf{0} \\ \mathbf{0} & \mathbf{0} & \mathbf{0} \end{pmatrix}$$

So,

$$A_{i,j}^{(1)} = \frac{(e_i W_Q^{(1)})(v_{i-j+1}^{(1)})^\top}{\sqrt{d}}.$$

Notice that $A_{i,j}^{(1)} = c\mathbb{1}[j = i - 1]$. So, $\text{softmax}(\text{mask}(A))_{i,j}^{(1)} \approx \mathbb{1}[j = i - 1]$ for large enough $c$. So, for any $2 \leq i < T, 1 \leq j < k$, $Attn_1(e)_{i,j+k} = e_{i-1,j}$. Effectively, the first layer appends the embedding of the previous token after the embedding of the current token, so that the output at position $i$ is approximately $(e_{x_i} \quad e_{x_{i-1}} \quad \mathbf{0})$.

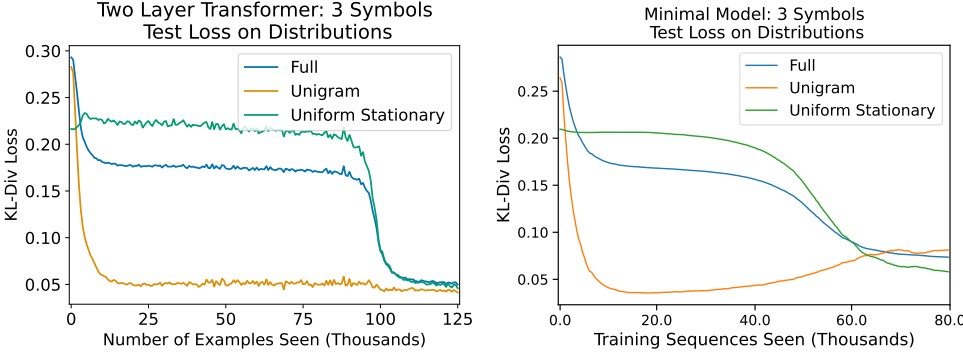

Figure 12: A two layer attention-only transformer (top) and minimal model (4) (bottom), trained on the main task with ICL-MC with cross entropy loss, test loss measured by KL-Divergence from the underlying truth (labels based on transition probabilities, not samples). The distributions test loss is measured in are (from left to right) in-distribution, a distribution where each token is sampled i.i.d., and a distribution over uniformly random doubly stochastic transition matrices (that is, a setting where the stationary distribution is the uniform distribution, which implies that unigram based guesses are as good as guessing uniform probability). For both models, the in distribution test loss quickly drops to the level of the unigram algorithm.

The second layer is defined as follows:

$$v^{(2)} = \mathbf{0} \quad W_Q^{(2)} = \begin{pmatrix} cI^{k\times k} & \mathbf{0} & \mathbf{0} \\ \mathbf{0} & \mathbf{0} & \mathbf{0} \\ \mathbf{0} & \mathbf{0} & \mathbf{0} \end{pmatrix} \quad W_K^{(2)} = \begin{pmatrix} \mathbf{0} & \mathbf{0} & \mathbf{0} \\ I^{k\times k} & \mathbf{0} & \mathbf{0} \\ \mathbf{0} & \mathbf{0} & \mathbf{0} \end{pmatrix} \quad W_V^{(2)} = \begin{pmatrix} \mathbf{0} & \mathbf{0} & I^{k\times k} \\ \mathbf{0} & \mathbf{0} & \mathbf{0} \\ \mathbf{0} & \mathbf{0} & \mathbf{0} \end{pmatrix}$$

Note that $z = e + Attn_1(e)$, then

$$A_{i,j}^{(2)} = \frac{(z_i W_Q^{(2)})(z_j W_K^{(2)})^\top}{\sqrt{d}} = \frac{ce_{x_i}(e_{x_{j-1}})^\top}{\sqrt{d}} = \frac{c}{\sqrt{d}}\mathbb{1}[x_{j-1} = x_i].$$

So, for all $j < i$, softmax$(\text{mask}(A))_{i,j} \approx \frac{\mathbb{1}[x_{j-1}=x_i]}{\sum_{h=1}^{i}\mathbb{1}[x_{h-1}=x_i]}$ for large enough $c$. For any $2 \leq i < T, 1 \leq j < k$,

$$Attn_2(e)_{i,j+2k} = \sum_{h=1}^{3k}\frac{\mathbb{1}[x_{h-1}=x_i]}{\sum_{g=1}^{i}\mathbb{1}[x_{g-1}=x_i]}(zW_V^{(2)})_{h,j} = \frac{\sum_{h=1}^{k}\mathbb{1}[x_{h-1}=x_i]\mathbb{1}[x_h=j]}{\sum_{g=1}^{i}\mathbb{1}[x_{g-1}=x_i]}.$$

This is exactly the empirical bigram statistics (that is, the number of times $x_i \to j$ appears before position $i$). In order to make this the output, we set $P = \begin{pmatrix} \mathbf{0} \\ \mathbf{0} \\ I^{k\times k} \end{pmatrix}$ [7]    □

## B.2    ICL-MC with Minimal Model

**Setup and notation**    Our data consists of sequences of length $T + 1$, $\boldsymbol{x} = (x_0, \ldots, x_T)$, drawn from a Markov Chain with state space $S = \{1, \ldots, k\}$ (i.e., $x_j \in \{1, \ldots, k\}$ for all $j \in [T]$), and a transition matrix $P$. Each row of the matrix is sampled from a flat Dirichlet distribution, i.e. $P_i \sim \text{Dir}(\mathbf{1})$, corresponding drawing the row from a uniform distribution over the simplex. Let

---

[7]Technically, the output of this construction is not the log probabilities as generally cross-entropy loss assumes. These can be approximated linearly by setting $P = \begin{pmatrix} b\mathbf{1}^\top\mathbf{1} \\ \mathbf{0} \\ aI^{k\times k} \end{pmatrix}$ to change the output from $x$ to $ax + b$. In practice, this approximation can achieve close to Bayes optimal loss.

$E \in \{0, 1\}^{(T+1) \times k}$ be the one hot embedding matrix of $x$, that is, $E_{i,x_i} = 1$ and for all $s \neq x_i$ $E_{i,s} = 0$. A crucial difference with parallel work which also studies in-context learning of Markov Chains [30] is that the whole sequence is sampled from the Markov Chain (whilst in [30], the penultimate token is sampled from the uniform distribution).

**Model** We define our model as a simplified sequence to sequence transformer $f : \mathbb{R}^{T \times k} \to \mathbb{R}^{(T+1) \times k}$ with $f(E) = \text{mask}(EW(\text{Softmax}(M)E)^\top)E$. The trainable parameters are $W \in \mathbb{R}^{k \times k}$ and $v \in \mathbb{R}^{T+1}$. We define $M \in \mathbb{R}^{(T+1) \times (T+1)}$ as $M = \begin{pmatrix} v_0 & -\infty & \dots & -\infty \\ v_1 & v_0 & \dots & -\infty \\ \vdots & \vdots & \dots & \vdots \\ v_T & v_{T-1} & \dots & v_0 \end{pmatrix}$, that is, for all $T \geq i \geq j \geq 0$, $M_{i,j} = v_{i-j}$ and if $i > j$, $M_{j,i} = -\infty$. Furthermore, $v = [v_0, v_2, \dots, v_T] \in \mathbb{R}^{t \times T+1}$. Softmax is defined as follows:

$$\text{Softmax}(M)_{i,j} = \frac{\exp(M_{i,j})}{\sum_{T=1}^{T} \exp(M_{i,j})}.$$

The logit for symbol $s$ at position $T$ for our model is:

$$f(E)_{T,s} = \sum_{u=1}^{k} \sum_{i=0}^{T} \sum_{j=0}^{i} W_{x_T,u} \mathbb{1}[x_{i-j} = u \wedge x_i = s] \frac{\exp(v_j)}{\sum_{\ell=0}^{i} \exp(v_\ell)}. \tag{5}$$

The model can represent the unigrams and bigrams solutions as following:

- Construction for bigrams: $v = (0, c, 0, \dots, 0)^\top$ and $W = I_{k \times k}$, then $f(E)_{T,s} = \sum_{i=0}^{T} \mathbb{1}[x_i = s \wedge x_{i-1} = x_T] + O\left(\frac{T^3}{\exp(c)}\right)$. As $c$ tends to infinity, this becomes bigrams.

- Construction for unigrams: $v = \mathbf{0}$ and $W = \frac{1}{k}\mathbf{1}^\top\mathbf{1}$, then $f(E)_{T,s} = \sum_{i=0}^{T} \mathbb{1}[x_i = s]$.

We prove the above claims.

**Proof of Fact 2.2**

*Proof.* We will first prove the unigrams construction.

$$f(E)_{T,s} = \sum_{u=1}^{k} \sum_{i=0}^{T} \sum_{j=0}^{i} W_{x_T,u} \mathbb{1}[x_{i-j} = u \wedge x_i = s] \frac{\exp(v_j)}{\sum_{\ell=0}^{i} \exp(v_\ell)}$$

$$= \sum_{u=1}^{k} \sum_{i=0}^{T} \sum_{j=0}^{i} \mathbb{1}[x_{i-j} = u \wedge x_i = s] \frac{1}{i}$$

$$= \frac{1}{k} \sum_{i=0}^{T} \sum_{j=0}^{i} \mathbb{1}[x_i = s] \frac{k}{i}$$

$$= \sum_{i=0}^{T} \mathbb{1}[x_i = s],$$

which is exactly the unigrams solution.

Now consider the bigrams construction. As $c$ grows, the softmax of $v$ very quickly becomes one hot. Formally, by Lemma B.7 in [17], for any $i > 0$,

$$\frac{\exp(v_j)}{\sum_{\ell=0}^{i} \exp(v_\ell)} = \mathbb{1}[j = 1] + O\left(\frac{T}{\exp(c)}\right)$$

So,

$$f(E)_{T,s} = \sum_{u=1}^{k}\sum_{i=0}^{T}\sum_{j=0}^{i} W_{x_T,u}\mathbb{1}[x_{i-j} = u \wedge x_i = s]\frac{\exp(v_j)}{\sum_{\ell=0}^{i}\exp(v_\ell)}$$

$$= \sum_{i=0}^{T}\sum_{j=0}^{i}\mathbb{1}[x_{i-j} = x_T \wedge x_i = s]\frac{\exp(v_j)}{\sum_{\ell=0}^{i}\exp(v_\ell)}$$

$$= \mathbb{1}[x_T = x_0 = s] + \sum_{i=1}^{T}\sum_{j=0}^{i}\mathbb{1}[x_T = x_{i-j} \wedge x_i = s]\frac{\exp(v_j)}{i - 1 + \exp(c))}$$

$$= \mathbb{1}[x_T = x_0 = s] + \sum_{i=1}^{T}\sum_{j=0}^{i}\mathbb{1}[x_T = x_{i-j} \wedge x_i = s]\left(\mathbb{1}[j=1] + O\left(\frac{2T}{\exp(c)}\right)\right) \quad \text{(Lemma B.7 in [17])}$$

$$= \mathbb{1}[x_T = x_0 = s] + \sum_{i=1}^{T}\mathbb{1}[x_T = x_{i-1} \wedge x_i = s] + \sum_{i=1}^{T}\sum_{j=0}^{i}\mathbb{1}[x_T = x_{i-j} \wedge x_i = s]O\left(\frac{2T}{\exp(c)}\right)$$

$$= \sum_{i=1}^{T}\mathbb{1}[x_T = x_{i-1} \wedge x_i = s] + \sum_{i=1}^{T}O\left(\frac{T^3}{\exp(c)}\right)$$

$\square$

This simplified model was constructed by taking a two layer transformer with relative positional encodings and simplifying it. Our construction for how transformers would form induction heads (corroborated with experiments such as the viewing of attention patterns in figure 2) implies that the MLPs and the value matrices could just be identity functions, and the first layer query matrix, and the second layer positional embeddings were zero matricies, so in the simplified model we froze these parameters to there final states. We also remove the softmax on the attention in the first layer. Despite these changes, the training dynamics, our main interest, stay remarkably similar.

**Training** We analyze gradient descent with the cross entropy loss $L_T(f, E, x_{T+1}) = -\sum_{s=1}^{k}\log \text{Softmax}\left(f(E)\right)_{T,s}P_{X_T,s}$[8]

### B.3   Gradient Calculations

For use in the proofs, here we show the calculations of the gradients of the model with respect to the parameters, and the loss with respect to the model.

$$\frac{\partial f(E)_{T,s}}{\partial W_{a,b}} = \sum_{u=1}^{k}\sum_{i=0}^{T}\sum_{j=0}^{i}\mathbb{1}[x_T = a \wedge b = u]\mathbb{1}[x_{i-j} = u \wedge x_i = s]\frac{\exp(v_j)}{\sum_{\ell=0}^{i}\exp(v_\ell)}$$

$$= \sum_{i=0}^{T}\sum_{j=0}^{i}\mathbb{1}[x_T = a]\mathbb{1}[x_{i-j} = b \wedge x_i = s]\frac{\exp(v_j)}{\sum_{\ell=0}^{i}\exp(v_\ell)}$$

$$\frac{\partial f(E)_{T,s}}{\partial v_a}$$

$$= \sum_{u=1}^{k}\sum_{i=0}^{T}\sum_{j=0}^{i} W_{x_T,u}\mathbb{1}[x_{i-j} = u \wedge x_i = s]\left(\mathbb{1}[j=a]\frac{\exp(v_a)}{\sum_{\ell=0}^{i}\exp(v_\ell)} - \mathbb{1}[a \le i]\frac{\exp(v_j)}{\sum_{\ell=0}^{i}\exp(v_\ell)}\frac{\exp(v_a)}{\sum_{\ell=0}^{i}\exp(v_\ell)}\right)$$

$$= \sum_{u=1}^{k}\sum_{i=0}^{T}\frac{\exp(v_a)}{\sum_{\ell=0}^{i}\exp(v_\ell)}\sum_{j=0}^{i} W_{x_T,u}\left(\mathbb{1}[x_{i-a} = u \wedge x_i = s]\mathbb{1}[j=a] - \mathbb{1}[x_{i-j} = u \wedge x_i = s]\mathbb{1}[a \le i]\frac{\exp(v_j)}{\sum_{\ell=0}^{i}\exp(v_\ell)}\right)$$

---

[8]In practice, one could use the empirical value of $x_{T+1}$ rather than its distribution $P_{X_T,s}$, but in full batch gradient descent this is in fact equivalent. This is because conditional on $x_T$ and $P$, $x_{T+1}$ is independent of $x_1, \ldots x_{T-1}$.

$$= \sum_{u=1}^{k} \sum_{i=0}^{T} \frac{\exp(v_a)}{\sum_{\ell=0}^{i} \exp(v_\ell)} W_{x_T, u} \left( \mathbb{1}[x_{i-a} = u \wedge x_i = s] \mathbb{1}[a \leq i] - \sum_{j=0}^{i} \mathbb{1}[x_{i-j} = u \wedge x_i = s] \mathbb{1}[a \leq i] \frac{\exp(v_j)}{\sum_{\ell=0}^{i} \exp(v_\ell)} \right)$$

$$= \sum_{u=1}^{k} \sum_{i=a}^{T} \frac{\exp(v_a)}{\sum_{\ell=0}^{i} \exp(v_\ell)} W_{x_T, u} \left( \mathbb{1}[x_{i-a} = u \wedge x_i = s] - \sum_{j=0}^{i} \mathbb{1}[x_{i-j} = u \wedge x_i = s] \frac{\exp(v_j)}{\sum_{\ell=0}^{i} \exp(v_\ell)} \right)$$

$$= \sum_{u=1}^{k} \sum_{i=a}^{T} \frac{\exp(v_a)}{\sum_{\ell=0}^{i} \exp(v_\ell)} W_{x_T, u} \mathbb{1}[x_i = s] \left( \mathbb{1}[x_{i-a} = u] - \sum_{j=0}^{i} \mathbb{1}[x_{i-j} = u] \frac{\exp(v_j)}{\sum_{\ell=0}^{i} \exp(v_\ell)} \right)$$

$$\frac{\partial L_T}{\partial f(E)_{T,s}} = \text{Softmax}(f(E))_{T,s} - P_{x_T, s}$$

## B.4 Proof of lemma 3.1

*Proof.* Recall that at initialization, $v = \mathbf{0}$ and $W = \mathbf{0}$, implying further that $f(E) = \mathbf{0}$.

**First step.**

First consider the gradient of the loss with respect to $W$. By chain rule,

$$\frac{\partial L_T(E)}{\partial W_{a,b}} = \sum_{s=1}^{k} \frac{\partial L_T}{\partial f(E)_{T,s}} \frac{\partial f(E)_{T,s}}{\partial W_{a,b}}$$

$$= \sum_{s=1}^{k} (\text{Softmax}(f(E))_{T,s} - P_{x_T, s}) \sum_{i=0}^{T} \sum_{j=0}^{i} \mathbb{1}[x_T = a] \mathbb{1}[x_{i-j} = b \wedge x_i = s] \frac{\exp(v_j)}{\sum_{\ell=0}^{i} \exp(v_\ell)}$$

$$= \sum_{s=1}^{k} \left( \frac{1}{k} - P_{x_T, s} \right) \mathbb{1}[x_T = a] \sum_{i=0}^{T} \sum_{j=0}^{i} \mathbb{1}[x_{i-j} = b \wedge x_i = s] \frac{1}{i+1}$$

$$= \frac{1}{k} \mathbb{1}[x_T = a] \sum_{i=0}^{T} \left( \sum_{j=0}^{i} \mathbb{1}[x_{i-j} = b] \frac{1}{i+1} - \sum_{s=1}^{k} P_{a,s} \mathbb{1}[x_T = a] \sum_{j=0}^{i} \mathbb{1}[x_{i-j} = b \wedge x_i = s] \frac{1}{i+1} \right)$$

$$= \frac{1}{k} \mathbb{1}[x_T = a] \sum_{i=0}^{T} \left( \mathbb{1}[x_i = b] - \sum_{s=1}^{k} P_{a,s} \mathbb{1}[x_T = a \sum_{j=0}^{i} \mathbb{1}[x_{i-j} = b \wedge x_i = s] \frac{1}{i+1} \right)$$

$$= \frac{1}{k} \sum_{i=0}^{T} \left( \mathbb{1}[x_i = b \wedge x_T = a] - \sum_{s=1}^{k} P_{a,s} \sum_{j=0}^{i} \mathbb{1}[x_{i-j} = b \wedge x_i = s \wedge x_T = a] \frac{1}{i+1} \right)$$

$$= \frac{1}{k} \sum_{i=0}^{T} \left( \mathbb{1}[x_0 = b \wedge x_{T-i} = a] - \sum_{s=1}^{k} P_{a,s} \sum_{j=0}^{i} \mathbb{1}[x_0 = b \wedge x_j = s \wedge x_{T-i+j} = a] \frac{1}{i+1} \right)$$

Where the last line follows from the Markov property.

Now we take the expectation over $x, x_{T+1}$ conditioned on the transition matrix $P$,

$$\mathbb{E}_{x|P} \left[ \frac{\partial L_T}{\partial W_{a,b}} \right] = \pi_b \sum_{i=0}^{T} \left( \frac{1}{k} \left( P^{T-i} \right)_{b,a} - \sum_{s=1}^{k} P_{a,s} \frac{1}{i+1} \left( P^{T-i} \right)_{s,a} \sum_{j=0}^{i} \left( P^j \right)_{b,s} \right)$$

$$= \pi_b \sum_{i=0}^{T} \left( \frac{1}{k} \left( P^i \right)_{b,a} - \sum_{s=1}^{k} P_{a,s} \frac{1}{T-i+1} \left( P^i \right)_{s,a} \sum_{j=0}^{T-i} \left( P^j \right)_{b,s} \right)$$

$$= \pi_b \pi_a (T+1) \left( \frac{1}{k} - \sum_{s=1}^{k} P_{a,s} \pi_s \right) + O(\log T)$$

Where the last step follows from Lemma B.6. Then, by applying Lemma B.1 (for the uniform over all $2 \times 2$ transition matrices case) or lemmas B.3 and B.4 (for the mixture of distributions case), there exist positive constants (potentially depending on $k$, but not $T$) $A, B$ such that for all $a$

$$\mathbb{E}_{x|P}\left[\frac{\partial L_T}{\partial W_{a,a}}\right] = -(A+B)T + O(\log T)$$

and for all $a \neq b$,

$$\mathbb{E}_{x|P}\left[\frac{\partial L_T}{\partial W_{a,b}}\right] = -BT + O(\log T)$$

The updated $W_{a,b}$ after the gradient step is just $-\eta_1 \mathbb{E}_{x|P}\left[\frac{\partial L_T}{\partial W_{a,b}}\right]$ (because $W$ is initialized at $\mathbf{0}$).

Choose $\eta_1 = \Theta\left(\frac{1}{T}\right)$, so that $W$ will be $O(1)$ with respect to $T$ after the first step.

For the gradient with respect to $v$, since $W = \mathbf{0}$,

$$\frac{\partial F(E)_{T,s}}{\partial v} = \sum_{u=1}^{k}\sum_{i=a}^{T} \frac{\exp(v_a)}{\sum_{\ell=0}^{i}\exp(v_\ell)} W_{x_T,u}\left(\mathbb{1}\left[x_{i-a} = u \wedge x_i = s\right] - \sum_{j=0}^{i}\mathbb{1}\left[x_{i-j} = u \wedge x_i = s\right]\frac{\exp(v_j)}{\sum_{\ell=0}^{i}\exp(v_\ell)}\right)$$

$$= 0$$

So,

$$\frac{\partial L_T(E)}{\partial v} = \sum_{s=1}^{k} \frac{\partial L_T(E)}{\partial f(E)_{T,s}} \frac{\partial F(E)_{T,s}}{\partial v} = 0$$

Completing the first step calculations.

**Second step.**

After the first step, $W = \eta_1\left(AI + B\mathbf{1}^\top\mathbf{1}\right)$. Now let us bound the output of the model,

$$|f(E)_{T,s}| = \left|\sum_{u=1}^{k}\sum_{i=0}^{T}\sum_{j=0}^{i} W_{x_T,u}\mathbb{1}\left[x_{i-j} = u \wedge x_i = s\right]\frac{\exp(v_j)}{\sum_{\ell=0}^{i}\exp(v_\ell)}\right|$$

$$= \left|\sum_{u=1}^{k}\sum_{i=0}^{T}\sum_{j=0}^{i} \eta_1\left(AI + B\mathbf{1}^\top\mathbf{1}\right)_{x_T,u}\mathbb{1}\left[x_{i-j} = u \wedge x_i = s\right]\frac{1}{i}\right|$$

$$\leq \eta_1\left|\sum_{u=1}^{k}\sum_{i=0}^{T}\sum_{j=0}^{i}(A+B)\,\mathbb{1}\left[x_{i-j} = u \wedge x_i = s\right]\frac{1}{i}\right|$$

$$\leq \eta_1\left|\sum_{u=1}^{k}\sum_{i=0}^{T}\sum_{j=0}^{i}(A+B)\,\mathbb{1}\left[x_{i-j} = u\right]\frac{1}{i}\right|$$

$$\leq \eta_1\left|\sum_{i=0}^{T}\sum_{j=0}^{i}(A+B)\frac{1}{i}\right|$$

$$\leq \eta_1 T\left|A+B\right|$$

So, using the first order approximation of softmax,

$$\frac{\partial L_T(E)}{\partial f(E)_{T,s}} = \text{Softmax}(f(E))_{T,s} - \mathbb{1}\left[x_{T+1} = s\right]$$

$$= \frac{1}{k} + \frac{f(E)_{T,s}}{k} - \frac{\sum_{u=1}^{k} f(E)_{T,u}}{k^2} + O(f(E)_{T,s}^2) - \mathbb{1}\left[x_{T+1} = s\right]$$

$$= \frac{1}{k} + O\left(\eta_1\frac{T}{k}(A+B)\right) + O(\eta_1^2 T^2(A+B)^2) - \mathbb{1}\left[x_{T+1} = s\right]$$

$$= \frac{1}{k} + O\left(\eta_1 \frac{T}{k}(A+B)\right) + O(\eta_1^2 T^2 (A+B)^2) - \mathbb{1}\left[x_{T+1} = s\right]$$

$$= \frac{1}{k} - \mathbb{1}\left[x_{T+1} = s\right] + O\left(\frac{1}{T}\right)$$

Where the last step follows since for the first step, $eta = O\left(\frac{1}{T^2}\right)$.

Now, we can begin to analyze the gradients with respect to the parameters. For $W$, the gradient is approximately the same as in the last step. Notice that $\frac{\partial f(E)_{T,s}}{\partial W_{a,b}}$ does not depend on $W$, and $v$ is unchanged, so $\frac{\partial f(E)_{T,s}}{\partial W_{a,b}}$ is unchanged. Furthermore,

$$\frac{\partial f(E)_{T,s}}{\partial W_{a,b}} = \sum_{s=1}^{k} \sum_{i=0}^{T} \sum_{j=0}^{i} \mathbb{1}\left[x_T = a\right]\mathbb{1}\left[x_{i-j} = b \wedge x_i = s\right] \frac{\exp(v_j)}{\sum_{\ell=0}^{i} \exp(v_\ell)}$$

$$= \sum_{i=0}^{T} \sum_{j=0}^{i} \mathbb{1}\left[x_T = a\right]\mathbb{1}\left[x_{i-j} = b\right] \frac{1}{i}$$

$$\leq \sum_{i=0}^{T} \sum_{j=0}^{i} \frac{1}{i}$$

$$= T$$

We will now show that the gradient is approximately the same as in the first gradient step:

$$\frac{\partial L_T(E)}{\partial W_{a,b}} = \sum_{s=1}^{k} \frac{\partial L_T}{\partial f(E)_{T,s}} \frac{\partial f(E)_{T,s}}{\partial W_{a,b}}$$

$$= \sum_{s=1}^{k} \left(\frac{1}{k} - \mathbb{1}\left[x_{T+1} = s\right] + O\left(\frac{1}{T}\right)\right) \frac{\partial f(E)_{T,s}}{\partial W_{a,b}}$$

$$= \sum_{s=1}^{k} \left(\frac{1}{k} - \mathbb{1}\left[x_{T+1} = s\right]\right) \frac{\partial f(E)_{T,s}}{\partial W_{a,b}} + O\left(\frac{1}{T}\right) \frac{\partial f(E)_{T,s}}{\partial W_{a,b}}$$

$$= \pi_b \pi_a (T+1) \left(\frac{1}{k} - \sum_{s=1}^{k} P_{a,s} \pi_s\right) + O(\log T)$$

Where the last lines follows from the gradient calculations in the first step.

Now we will consider the gradient with respect to $v$. First, notice that the uniform component of $W$, $B\mathbf{1}^\top\mathbf{1}$, has no affect on the gradient of $v$:

$$\frac{\partial f(E)_{T,s}}{\partial v_a} = \sum_{u=1}^{k} \sum_{i=a}^{T} W_{x_T,u} \frac{\exp(v_a)}{\sum_{\ell=0}^{i} \exp(v_\ell)} \mathbb{1}\left[x_i = s\right] \left(\mathbb{1}\left[x_{i-a} = u\right] - \sum_{j=0}^{i} \frac{\exp(v_j)}{\sum_{\ell=0}^{i} \exp(v_\ell)} \mathbb{1}\left[x_{i-j} = u\right]\right)$$

$$= \sum_{u=1}^{k} \sum_{i=a}^{T} \left(mI + B\mathbf{1}^\top\mathbf{1}\right)_{x_T,u} \frac{1}{i+1} \mathbb{1}\left[x_i = s\right] \left(\mathbb{1}\left[x_{i-a} = u\right] - \sum_{j=0}^{i} \frac{1}{i+1} \mathbb{1}\left[x_{i-j} = u\right]\right)$$

$$= \sum_{u=1}^{k} \sum_{i=a}^{T} \left(A\mathbb{1}\left[x_T = u\right] + B\right) \frac{1}{i+1} \mathbb{1}\left[x_i = s\right] \left(\mathbb{1}\left[x_{i-a} = u\right] - \sum_{j=0}^{i} \frac{1}{i+1} \mathbb{1}\left[x_{i-j} = u\right]\right)$$

$$= A \sum_{u=1}^{k} \sum_{i=a}^{T} \mathbb{1}\left[x_T = u\right] \frac{1}{i+1} \mathbb{1}\left[x_i = s\right] \left(\mathbb{1}\left[x_{i-a} = u\right] - \sum_{j=0}^{i} \frac{1}{i+1} \mathbb{1}\left[x_{i-j} = u\right]\right)$$

$$+ B \sum_{u=1}^{k} \sum_{i=a}^{T} \frac{1}{i+1} \mathbb{1}\left[x_i = s\right] \left(\mathbb{1}\left[x_{i-a} = u\right] - \sum_{j=0}^{i} \frac{1}{i+1} \mathbb{1}\left[x_{i-j} = u\right]\right)$$

$$= A \sum_{u=1}^{k} \sum_{i=a}^{T} \mathbb{1}[x_T = u] \frac{1}{i+1} \mathbb{1}[x_i = s] \left( \mathbb{1}[x_{i-a} = u] - \sum_{j=0}^{i} \frac{1}{i+1} \mathbb{1}[x_{i-j} = u] \right)$$

$$+ B \sum_{i=a}^{T} \frac{1}{i+1} \mathbb{1}[x_i = s] \left( \sum_{u=1}^{k} \mathbb{1}[x_{i-a} = u] - \sum_{j=0}^{i} \frac{1}{i+1} \sum_{u=1}^{k} \mathbb{1}[x_{i-j} = u] \right)$$

$$= A \sum_{u=1}^{k} \sum_{i=a}^{T} \mathbb{1}[x_T = u] \frac{1}{i+1} \mathbb{1}[x_i = s] \left( \mathbb{1}[x_{i-a} = u] - \sum_{j=0}^{i} \frac{1}{i+1} \mathbb{1}[x_{i-j} = u] \right)$$

$$+ B \sum_{i=a}^{T} \frac{1}{i+1} \mathbb{1}[x_i = s] \left( 1 - \sum_{j=0}^{i} \frac{1}{i+1} \right)$$

$$= A \sum_{u=1}^{k} \sum_{i=a}^{T} \mathbb{1}[x_T = u] \frac{1}{i+1} \mathbb{1}[x_i = s] \left( \mathbb{1}[x_{i-a} = u] - \sum_{j=0}^{i} \frac{1}{i+1} \mathbb{1}[x_{i-j} = u] \right)$$

By chain rule,

$$\frac{\partial L_T}{\partial v_a} = \sum_{s=1}^{k} \frac{\partial L_T}{\partial f(E)_{T,s}} \frac{\partial f(E)_{T,s}}{\partial v_a}$$

$$= \sum_{s=1}^{k} \left( \frac{1}{k} - P_{x_T,s} + O\left(\frac{1}{T}\right) \right) \sum_{u=1}^{k} \sum_{i=a}^{T} \mathbb{1}[x_T = u] \frac{1}{i+1} \mathbb{1}[x_i = s] \left( \mathbb{1}[x_{i-a} = u] - \sum_{j=0}^{i} \frac{1}{i+1} \mathbb{1}[x_{i-j} = u] \right)$$

$$= \sum_{s=1}^{k} \left( \frac{1}{k} - P_{x_T,s} \right) \sum_{u=1}^{k} \sum_{i=a}^{T} \mathbb{1}[x_T = u] \frac{1}{i+1} \mathbb{1}[x_i = s] \left( \mathbb{1}[x_{i-a} = u] - \sum_{j=0}^{i} \frac{1}{i+1} \mathbb{1}[x_{i-j} = u] \right) + O\left(\frac{\log T}{T}\right)$$

Where the last step follows because

$$\left| \sum_{u=1}^{k} \sum_{i=a}^{T} \mathbb{1}[x_T = u] \frac{1}{i+1} \mathbb{1}[x_i = s] \left( \mathbb{1}[x_{i-a} = u] - \sum_{j=0}^{i} \frac{1}{i+1} \mathbb{1}[x_{i-j} = u] \right) \right| \leq \left| \sum_{u=1}^{k} \sum_{i=a}^{T} \mathbb{1}[x_T = u] \frac{1}{i+1} \right|$$

$$= \left| \sum_{i=a}^{T} \frac{1}{i+1} \right|$$

$$\leq \log T$$

In expectation over the values of $x$, conditioned on the choice of $P$:

$$\mathbb{E}_{x|P} \left[ \frac{\partial L_T}{\partial v_a} \right] = \sum_{s=1}^{k} \sum_{u=1}^{k} \left( \frac{1}{k} - P_{u,s} \right) \sum_{i=a}^{T} \frac{\pi_u}{i+1} \left( P^{T-i} \right)_{s,u} \left( \left( P^a \right)_{u,s} - \frac{1}{i+1} \sum_{j=0}^{i} \left( P^{i-j} \right)_{u,s} \right) + O\left(\frac{\log T}{T}\right)$$

$$= \sum_{s=1}^{k} \sum_{u=1}^{k} \left( \frac{1}{k} - P_{u,s} \right) \sum_{i=a}^{T} \frac{\pi_u}{T-i+1} \left( P^i \right)_{s,u} \left( \left( P^a \right)_{u,s} - \frac{1}{T-i+1} \sum_{j=0}^{T-i} \left( P^j \right)_{u,s} \right) + O\left(\frac{\log T}{T}\right)$$

$$= \left( \log(T+1) - \log(a+1) \right) \sum_{s=1}^{k} \sum_{u=1}^{k} \pi_u^2 P_{u,s} \left( \pi_s - \left( P^a \right)_{u,s} \right) + O(1)$$

Where the last step follows from Lemma B.7. Then, by applying Lemma B.2 or lemmas B.3 and B.4 (depending on the distribution assumption on $P$),

$$\mathbb{E}_{x|P} \left[ \frac{\partial L_T}{\partial v_1} \right] < \mathbb{E}_{x|P} \left[ \frac{\partial L_T}{\partial v_a} \right]$$

$$\mathbb{E}_{x|P}\left[\frac{\partial L_T}{\partial v_1}\right] < 0$$

Therefore, after the step is taken,

$$v_1 = \Theta(\eta_2 \log T)$$
$$v_1 - v_n = \eta_2 \Omega(\log T)$$

Finally, we can consider the state of the model after the second step. Assume that the step size for $v$ in the second step is $O(T)$, and the step size for $W$ is $\frac{1}{T(A+B)}$

$$f(E)_{T,s} = \sum_{u=1}^{k}\sum_{i=0}^{T}\sum_{j=0}^{i} W_{x_T,u}\mathbb{1}[x_{i-j}=u \wedge x_i = s]\frac{\exp(v_j)}{\sum_{\ell=0}^{i}\exp(v_\ell)}$$

$$= \frac{1}{A+B}\sum_{u=1}^{k}\sum_{i=0}^{T}\sum_{j=0}^{i}\left(AI + B\mathbf{1}^\top\mathbf{1} + O\left(\frac{\log T}{T}\right)\right)_{x_T,u}\mathbb{1}[x_{i-j}=u \wedge x_i = s]\left(\mathbb{1}[j=1] + O\left(\frac{2T}{\exp(\log(T))}\right)\right)$$

$$= \frac{1}{A+B}\sum_{u=1}^{k}\sum_{i=0}^{T}\sum_{j=0}^{i}\left(AI + B\mathbf{1}^\top\mathbf{1} + O\left(\frac{\log T}{T}\right)\right)_{x_T,u}\mathbb{1}[x_{i-j}=u \wedge x_i = s]\left(\mathbb{1}[j=1] + O\left(\frac{1}{T}\right)\right)$$

$$= \frac{1}{A+B}\sum_{u=1}^{k}\sum_{i=0}^{T}\left(AI + B\mathbf{1}^\top\mathbf{1}\right)_{x_T,u}\mathbb{1}[x_{i-1}=u \wedge x_i = s] + O(\log T)$$

$$= \frac{A}{A+B}\sum_{i=0}^{T}\mathbb{1}[x_{i-1}=x_T \wedge x_i = s] + \frac{B}{A+B}\sum_{u=1}^{k}\sum_{i=0}^{T}\mathbb{1}[x_{i-1}=u \wedge x_i = s] + O(\log T)$$

$$= \frac{A}{A+B}\sum_{i=0}^{T}\mathbb{1}[x_{i-1}=x_T \wedge x_i = s] + \frac{B}{A+B}\sum_{i=0}^{T}\mathbb{1}[x_i = s] + O(\log T)$$

Since $A$ and $B$ are constant in terms of $T$, we can recover the desired statements, completing the proof. $\qquad\square$

## B.5 Inequality lemmas for $k = 2$

**Lemma B.1.** *If $P$ is a uniformly random stochastic $2 \times 2$ matrix, and $\pi$ is the stationary distribution of $P$, then*

$$\mathbb{E}\left[\pi_a^2\left(\frac{1}{k} - \sum_{s=1}^{k}P_{a,s}\pi_s\right)\right] = \frac{5}{12} - \frac{2}{3}\log(2) \approx -0.045$$

*and for any $b \neq a$*

$$\mathbb{E}\left[\pi_a\pi_b\left(\frac{1}{k} - \sum_{s=1}^{k}P_{a,s}\pi_s\right)\right] = -\frac{7}{6} + \frac{5}{3}\log(2) \approx -0.011$$

*Proof.* We have:

$$\mathbb{E}\left[\pi_a^2\left(\frac{1}{k} - \sum_{s=1}^{k}P_{a,s}\pi_s\right)\right] = \mathbb{E}_{a,b}\left[\frac{(b-1)^2}{(a+b-2)^2}\left[\frac{1}{2} - \frac{a(b-1)}{a+b-2} - \frac{(1-a)(a-1)}{a+b-2}\right]\right]$$

$$= \frac{1}{2}\int_0^1\int_0^1\frac{(b-1)^2}{(a+b-2)^2}dadb - \int_0^1\int_0^1\frac{a(b-1)^3}{(a+b-2)^3}dadb + \int_0^1\int_0^1\frac{(b-1)^2(a-1)^2}{(a+b-2)^3}dadb$$

$$= \frac{1}{2}(1-\ln 2) - \frac{1}{2}(1-\ln 2) + \frac{5}{12}(5-8\ln 2) = \frac{5}{12} - \frac{2}{3}\ln 2.$$

$$(6)$$

For the non-diagonal elements, it holds:

$$\mathbb{E}\left[\pi_a \pi_b \left(\frac{1}{k} - \sum_{s=1}^{k} P_{a,s} \pi_s\right)\right]$$

$$= \frac{1}{2} \int_0^1 \int_0^1 \frac{(b-1)(a-1)}{(a+b-2)^2} \, da \, db - \int_0^1 \int_0^1 \frac{a(b-1)^2(a-1)}{(a+b-2)^3} \, da \, db + \int_0^1 \int_0^1 \frac{(b-1)(a-1)^3}{(a+b-2)^3} \, da \, db$$

$$= \frac{1}{2}\left(\ln 2 - \frac{1}{2}\right) - \frac{1}{6}(1 - \ln 2) + \left(\ln 2 - \frac{3}{4}\right) = \frac{5}{3}\ln 2 - \frac{7}{6}.$$

$$(7)$$

$\square$

**Lemma B.2.** *If $P$ is a uniformly random stochastic $2 \times 2$ matrix, and $\pi$ is the stationary distribution of $P$, then,*

$$\mathbb{E}\left[\sum_{s=1}^{k}\sum_{u=1}^{k} \pi_u^2 P_{u,s}(\pi_s - P_{u,s})\right] = -7/2 + 5\log(2) \approx -0.034$$

*and for any $n \neq 1$*

$$\mathbb{E}\left[\sum_{s=1}^{k}\sum_{u=1}^{k} \pi_u^2 P_{u,s}(\pi_s - P_{u,s})\right] \leq \mathbb{E}\left[\sum_{s=1}^{k}\sum_{u=1}^{k} \pi_u^2 P_{u,s}\left(\pi_s - (P^n)_{u,s}\right)\right]$$

*Proof.* We have:

$$\mathbb{E}\left[\sum_{s=1}^{k}\sum_{u=1}^{k} \pi_u^2 P_{u,s}(\pi_s - P_{u,s})\right] =$$

$$\left[\frac{1/6 x^4 + (x+y)(6xy(-4x^2 + 2x + 1) + 6y^4 + y^3(20 - 24x)}{12(x+y)}\right.$$

$$\left. + \frac{y^2(12x^2 - 12x - 3) + \log((x+y)^{6x^2(4x^2+2x-1)}(x+y)^{6y^2(4y^2+2y-1)}))}{12(x+y)}\right]_0^1$$

$$(8)$$

$$= -7/2 + 5\log(2)$$

For the inequality, we have an intuition that doesn't depend on $k$, notice that:

$$\sum_{s=1}^{k}\sum_{u=1}^{k} \pi_u^2 P_{u,s}\left(\pi_s - (P^n)_{u,s}\right) \geq -\sum_{s=1}^{k}\sum_{u=1}^{k} \pi_u^2 P_{u,s}\left|\pi_s - (P^a)_{u,s}\right|$$

$$\geq -\sum_{s=1}^{k}\sum_{u=1}^{k} \pi_u^2 P_{u,s} \alpha^n$$

$$= -\sum_{u=1}^{k} \pi_u^2 \alpha^n$$

$$\geq -\alpha^n$$

As long as $\alpha$ isn't concentrated around 1, then this shows that the magnitude of the RHS is bounded by a term that shrinks exponentially in $n$. For $k = 2$, we will find a similar bound, and then show separately that for all $n$ for which the bound fails, the inequality still holds true.

$$\sum_{s=1}^{k}\sum_{u=1}^{k} \pi_u^2 P_{u,s}\left(\pi_s - (P^n)_{u,s}\right) = \frac{P_{1,2}P_{2,1}(4P_{1,2}P_{2,1} - P_{1,2} - P_{2,1})}{(P_{1,2} + P_{2,1})^3}(1 - P_{1,2} - P_{2,1})^n$$

We can show that for any choice of $P_{1,2}$ and $P_{2,1}$ on the unit square,

$$\left|\frac{P_{1,2}P_{2,1}(4P_{1,2}P_{2,1} - P_{1,2} - P_{2,1})}{(P_{1,2} + P_{2,1})^3}\right| \leq \frac{1}{4}$$

To see why this is true, observe that,

$$(4P_{1,2}P_{2,1} - P_{1,2} - P_{2,1})^2$$
$$= 16P_{1,2}^2P_{2,1}^2 + (P_{1,2} + P_{2,1})^2 - 8(P_{1,2} + P_{2,1})P_{1,2}P_{2,1}$$
$$\leq 16P_{1,2}^2P_{2,1}^2 + (P_{1,2} + P_{2,1})^2 - 4(P_{1,2} + P_{2,1})^2P_{1,2}P_{2,1} \qquad \text{since } P_{1,2} + P_{2,1} \leq 2$$
$$= 16P_{1,2}^2P_{2,1}^2 + (P_{1,2} + P_{2,1})^2 - 4P_{1,2}P_{2,1}((P_{1,2} + P_{2,1})^2 - 4P_{1,2}P_{2,1})$$
$$= (P_{1,2} + P_{2,1})^2 - 4P_{1,2}P_{2,1}(P_{1,2} - P_{2,1})^2$$
$$\leq (P_{1,2} + P_{2,1})^2$$

Using the above, we have

$$\left( \frac{P_{1,2}P_{2,1}(4P_{1,2}P_{2,1} - P_{1,2} - P_{2,1})}{(P_{1,2} + P_{2,1})^3} \right)^2 \leq \frac{P_{1,2}^2P_{2,1}^2(P_{1,2} + P_{2,1})^2}{(P_{1,2} + P_{2,1})^6}$$
$$= \frac{P_{1,2}^2P_{2,1}^2}{(P_{1,2} + P_{2,1})^4}$$
$$\leq \frac{P_{1,2}^2P_{2,1}^2}{16P_{1,2}^2P_{2,1}^2} \qquad \text{using } (P_{1,2} + P_{2,1})^2 \geq 4P_{1,2}P_{2,1}$$
$$= \frac{1}{16}.$$

So,

$$\| \sum_{s=1}^{k} \sum_{u=1}^{k} \pi_u^2 P_{u,s} \left( \pi_s - (P^n)_{u,s} \right) \| \leq \frac{1}{4} |1 - P_{1,2} - P_{2,1}|^n$$

Now,

$$\mathbb{E}\left[ -1\frac{1}{4} |1 - P_{1,2} - P_{2,1}|^n \right] = -\frac{1}{4} \int_0^1 \int_0^1 |1 - x - y|^n$$
$$= -\frac{1}{4} \frac{2}{(n+1)(n+2)}$$
$$= -\frac{1}{2(n+1)(n+2)}$$

Notice that this decreases in $n$, and at $n = 3$, $\frac{1}{2(3+1)(3+2)} = \frac{1}{40} = 0.025$ which is less in magnitude than the value we proved at $n = 1$, $|-7/2 + 5\log 2| \approx 0.034$. So, solving for $n = 2$ (verified by a symbolic algebra program)

$$\mathbb{E}\left[ \frac{P_{1,2}P_{2,1} \left( -P_{1,2} - P_{2,1} + 1 \right)^2 \cdot (2P_{1,2}P_{2,1} + P_{1,2} \left( P_{2,1} - 1 \right) + P_{2,1} \left( P_{1,2} - 1 \right))}{(P_{1,2} + P_{2,1})^3} \right] = -\frac{413}{60} + \frac{149 \log (2)}{15} \approx 0.002$$

Which is not only greater than $-7/2 + 5\log 2$, but positive. Lastly, we simply need to show that the inequality holds at $n = 0$, and we are done.

$$\mathbb{E}\left[ \frac{P_{1,2}P_{2,1} \left( -P_{1,2} - P_{2,1} + 1 \right)^0 \cdot (2P_{1,2}P_{2,1} + P_{1,2} \left( P_{2,1} - 1 \right) + P_{2,1} \left( P_{1,2} - 1 \right))}{(P_{1,2} + P_{2,1})^3} \right]$$
$$= -\mathbb{E}\left[ \frac{P_{1,2}P_{2,1} \cdot (2P_{1,2}P_{2,1} + P_{1,2} \left( P_{2,1} - 1 \right) + P_{2,1} \left( P_{1,2} - 1 \right))}{(P_{1,2} + P_{2,1})^3} \right]$$
$$= -7/6 + 5 * log(2)/3 \approx -0.0114$$

Which is greater than $-7/2 + 5\log 2$, completing our proof. □

**Lemma B.3.** *If $P$ is a uniformly random doubly stochastic matrix, then,*

$$\mathbb{E}\left[\pi_a^2\left(\frac{1}{k} - \sum_{s=1}^{k} P_{a,s}\pi_s\right)\right] = \mathbb{E}\left[\pi_a\pi_b\left(\frac{1}{k} - \sum_{s=1}^{k} P_{a,s}\pi_s\right)\right]$$

*for all $a, b$ and*

$$\mathbb{E}\left[\sum_{s=1}^{k}\sum_{u=1}^{k}\pi_u^2 P_{u,s}\left(\pi_s - P_{u,s}\right)\right] < \mathbb{E}\left[\sum_{s=1}^{k}\sum_{u=1}^{k}\pi_u^2 P_{u,s}\left(\pi_s - (P^a)_{u,s}\right)\right]$$

*For all non-negative $a \neq 1$. and*

$$\mathbb{E}\left[\sum_{s=1}^{k}\sum_{u=1}^{k}\pi_u^2 P_{u,s}\left(\pi_s - P_{u,s}\right)\right] < 0$$

*Proof.* We will use the fact that for doubly stochastic matrices, the stationary distribution is the uniform vector $\frac{1}{k}\mathbf{1}$.

The first equality follows directly from $\pi_a = \frac{1}{k} = \pi_b$. Now we will prove the first inequality.

$$\mathbb{E}\left[\sum_{s=1}^{k}\sum_{u=1}^{k}\pi_u^2 P_{u,s}\left(\pi_s - (P^a)_{u,s}\right)\right] = \mathbb{E}\left[\sum_{s=1}^{k}\sum_{u=1}^{k}\frac{1}{k^2}P_{u,s}\left(\frac{1}{k} - (P^a)_{u,s}\right)\right]$$

$$= \frac{1}{k^2} - \frac{1}{k^2}\sum_{s=1}^{k}\sum_{u=1}^{k}\mathbb{E}\left[P_{u,s}(P^a)_{u,s}\right]$$

$$= \frac{1}{k^2} - \frac{1}{k^2}\mathbb{E}\left[\langle P, P^a\rangle_F\right]$$

Where $\langle .,.\rangle_F$ is the Frobenius inner product. We will first consider the case where $a > 1$. Notice that it is sufficient to prove that for any doubly stochastic $P$ (excluding the measure zero case of where $P = P^2$, where equality is reached), $\langle P, P^a\rangle_F < \|P\|_F^2$. First, by Cauchy–Schwarz,

$$\langle P, P^a\rangle_F < \|P\|_F\|P^a\|_F$$

We can use strictly less than because Cauchy Schwarz is only tight when $P$ and $P^a$ are linearly dependent, and since both $P$ and $P^a$ are doubly stochastic, linear dependence implies equality, which is only the case when $P = P^a$. Then, For now assume $a > 0$, then,

$$= \|P\|_F\|P^a\|_F$$

$$= \|P\|_F\|PP^{a-1}\|_F$$

$$= \|P\|_F\|\sum_i \alpha_i\Lambda_i P^{a-1}\|_F$$

$$\leq \sum_i \alpha_i\|P\|_F\|\Lambda_i P^{a-1}\|_F$$

$$= \sum_i \alpha_i\|P\|_F\|P^{a-1}\|_F$$

$$= \|P\|_F\|P^{a-1}\|_F$$

The third step used the well known Birkhoff-Von Neumann Theorem [9] that any doubly stochastic matrix $P$ is the convex combination of permutation matrices, so $P = \sum_i \alpha_i\Lambda_i$ for some permutation matrices $\Lambda_i$ and constants $\alpha_i > 0$ with $\sum_i \alpha_i = 1$. The inequality step uses the triangle inequality. Induction on positive $a$ yields the desired inequality for positive $a$.

Now consider the remaining case, $a = 0$,

$$\frac{1}{k^2} - \frac{1}{k^2}\sum_{s=1}^{k}\sum_{u=1}^{k}\mathbb{E}\left[P_{u,s}(P^0)_{u,s}\right] = \frac{1}{k^2} - \frac{1}{k^2}\sum_{s=1}^{k}\mathbb{E}\left[P_{s,s}\right]$$

$$= \frac{1}{k^2} - \frac{1}{k^2} = 0.$$

While at $a = 1$, for any $P$ that isn't $\mathbb{1}\mathbb{1}^\top$, $\|P\|_F > 1$, so

$$\frac{1}{k^2} - \frac{1}{k^2} \sum_{s=1}^{k} \sum_{u=1}^{k} \mathbb{E}\left[P_{u,s}^2\right] < \frac{1}{k^2} - \frac{1}{k^2} = 0.$$

Completing the proof of both inequalities.

$\square$

**Lemma B.4.** *If $P$ is a uniformly random $k \times k$ stochastic matrix subject to each row being the same, then,*

$$\mathbb{E}\left[\pi_a^2 \left(\frac{1}{k} - \sum_{s=1}^{k} P_{a,s}\pi_s\right)\right] < \mathbb{E}\left[\pi_a\pi_b \left(\frac{1}{k} - \sum_{s=1}^{k} P_{a,s}\pi_s\right)\right] < 0$$

*and*

$$\frac{\mathbb{E}\left[\pi_a^2 \left(\frac{1}{k} - \sum_{s=1}^{k} P_{a,s}\pi_s\right)\right]}{\mathbb{E}\left[\pi_a\pi_b \left(\frac{1}{k} - \sum_{s=1}^{k} P_{a,s}\pi_s\right)\right]} \geq \frac{8}{5}$$

*for all $a$ and $b$ and*

$$\mathbb{E}\left[\sum_{s=1}^{k} \sum_{u=1}^{k} \pi_u^2 P_{u,s} \left(\pi_s - (P^a)_{u,s}\right)\right] = 0$$

*For all $a$.*

*Proof.* The equality statement follows from the facts that for such transition matrices, $P^a = P$ for all natural $a > 0$, and that the stationary distribution matches the rows, that is, for any $a, b, \pi_b = P_{a,b}$,

$$\mathbb{E}\left[\sum_{s=1}^{k} \sum_{u=1}^{k} \pi_u^2 P_{u,s} \left(\pi_s - (P^a)_{u,s}\right)\right] = \mathbb{E}\left[\sum_{s=1}^{k} \sum_{u=1}^{k} \pi_u^2 P_{u,s} \left(\pi_s - P_{u,s}\right)\right] = \mathbb{E}\left[\sum_{s=1}^{k} \sum_{u=1}^{k} \pi_u^2 P_{u,s} \left(\pi_s - \pi_s\right)\right] = 0$$

Now we will do the inequalities. We will also use the following facts derived from the moments of the Dirichlet distribution,

$$E\left[\|\pi\|_2^2\right] = \frac{2}{k+1}$$

$$E\left[\|\pi\|_2^4\right] = \frac{4(k+5)}{(k+1)(k+2)(k+3)}$$

So,

$$\mathbb{E}\left[\pi_a^2 \left(\frac{1}{k} - \sum_{s=1}^{k} P_{a,s}\pi_s\right)\right] = \mathbb{E}\left[\pi_a^2 \left(\frac{1}{k} - \sum_{s=1}^{k} \pi_s^2\right)\right]$$

$$= \frac{1}{k}\mathbb{E}\left[\|\pi\|_2^2 \left(\frac{1}{k} - \|\pi\|_2^2\right)\right]$$

$$= \frac{1}{k^2}\mathbb{E}\left[\|\pi\|_2^2\right] - \frac{1}{k}\mathbb{E}\left[\left(\|\pi\|_2^4\right)\right]$$

$$= \frac{2}{k^2(k+1)} - \frac{4(k+5)}{k(k+1)(k+2)(k+3)}$$

Which is negative for all $k \geq 2$. And,

$$\mathbb{E}\left[\pi_a\pi_b \left(\frac{1}{k} - \sum_{s=1}^{k} P_{a,s}\pi_s\right)\right] = \mathbb{E}\left[\pi_a\pi_b \left(\frac{1}{k} - \sum_{s=1}^{k} \pi_s^2\right)\right]$$

$$= \frac{1}{k^2} \mathbb{E}\left[\left(\frac{1}{k} - \|\pi\|_2^2\right)\right]$$

$$= \frac{1}{k^3} - \frac{1}{k^2} \mathbb{E}\left[\|\pi\|_2^2\right]$$

$$= \frac{1}{k^3} - \frac{2}{k^2(k+1)}$$

Which is also negative for all $k \geq 2$. Finally, notice that

$$\frac{\frac{2}{k^2(k+1)} - \frac{4(k+5)}{k(k+1)(k+2)(k+3)}}{\frac{1}{k^3} - \frac{2}{k^2(k+1)}} \geq \frac{8}{5}$$

For all $k \geq 2$. $\qquad\square$

## B.6 Approximation Lemmata

The following lemma is a well known property of stochastic matrices, (see Lemma 3.3.2 Gallager [19] for example).

**Lemma B.5.** *Let* $\alpha = 1 - 2\min_{i,j} P_{i,j}$. *Then, for any* $i, j$

$$\left|(P^n)_{i,j} - \pi_j\right| \leq \alpha^n$$

Lemma B.6 and Lemma B.7 both share similar intuitions and proofs. They largely rely on Lemma B.5, which shows that $(P^n)_{i,j}$ approaches $\pi_j$ exponentially fast with respect to $n$, to show that over the course of summations over $n$ the stationary distribution dominates, allowing us to simplify the expressions.

**Lemma B.6.** *Let* $P$ *be a stochastic matrix with all positive entries, and let* $a, b$ *be states. Assume that* $\min_{i,j} P_{i,j}$ *is positive and doesn't dependend on* $T$. *Then,*

$$\pi_b \sum_{i=0}^{T} \left(\frac{1}{k}\left(P^i\right)_{b,a} - \sum_{s=1}^{k} P_{a,s} \frac{1}{T-i+1} \left(P^i\right)_{s,a} \sum_{j=0}^{T-i} \left(P^j\right)_{b,s}\right)$$

$$= \pi_b \pi_a (T+1) \left(\frac{1}{k} - \sum_{s=1}^{k} P_{a,s}\pi_s\right) + O(\log T).$$

*Proof.* Let us bound the magnitude of the difference between the two expressions.

$$\left|\pi_b \sum_{i=0}^{T} \left(\frac{1}{k}\left(P^i\right)_{b,a} - \sum_{s=1}^{k} P_{a,s} \frac{1}{T-i+1} \left(P^i\right)_{s,a} \sum_{j=0}^{T-i} \left(P^j\right)_{b,s}\right) - \pi_b \pi_a (T+1) \left(\frac{1}{k} - \sum_{s=1}^{k} P_{a,s}\pi_s\right)\right|$$

$$= \left|\pi_b \sum_{i=0}^{T} \left(\frac{1}{k}\left(\left(P^i\right)_{b,a} - \pi_a\right) - \sum_{s=1}^{k} P_{a,s} \left(\frac{1}{T-i+1} \left(P^i\right)_{s,a} \sum_{j=0}^{T-i} \left(P^j\right)_{b,s} - \pi_s \pi_a\right)\right)\right|$$

$$\leq \pi_b \sum_{i=0}^{T} \left(\frac{1}{k}\left|\left(P^i\right)_{b,a} - \pi_a\right| + \sum_{s=1}^{k} P_{a,s} \frac{1}{T-i+1} \sum_{j=0}^{T-i} \left|\left(P^i\right)_{s,a} \left(P^j\right)_{b,s} - \pi_s \pi_a\right|\right)$$

$$\leq \pi_b \sum_{i=0}^{T} \left(\frac{1}{k}\alpha^i + \sum_{s=1}^{k} P_{a,s} \frac{1}{T-i+1} \sum_{j=0}^{T-i} \left|\left(P^i\right)_{s,a} \left(P^j\right)_{b,s} - \pi_s \pi_a\right|\right)$$

$$= \pi_b \sum_{i=0}^{T} \left(\frac{1}{k}\alpha^i + \sum_{s=1}^{k} P_{a,s} \frac{1}{T-i+1} \sum_{j=0}^{T-i} \left|\left(\left(P^j\right)_{b,s} \left(P^i\right)_{s,a} - \pi_a\right) + \pi_a\left(\left(P^j\right)_{b,s} - \pi_s\right)\right|\right)$$

$$\leq \pi_b \sum_{i=0}^{T} \left(\frac{1}{k}\alpha^i + \sum_{s=1}^{k} P_{a,s} \frac{1}{T-i+1} \sum_{j=0}^{T-i} \left(\left(P^j\right)_{b,s} \left|\left(P^i\right)_{s,a} - \pi_a\right| + \pi_a \left|\left(P^j\right)_{b,s} - \pi_s\right|\right)\right)$$

$$\leq \pi_b \sum_{i=0}^{T} \left( \frac{1}{k} \alpha^i + \sum_{s=1}^{k} P_{a,s} \frac{1}{T-i+1} \sum_{j=0}^{T-i} \left( (P^j)_{b,s} \alpha^i + \pi_a \alpha^j \right) \right)$$

$$\leq \pi_b \sum_{i=0}^{T} \left( \frac{1}{k} \alpha^i + \sum_{s=1}^{k} P_{a,s} \frac{1}{T-i+1} \sum_{j=0}^{T-i} \left( \alpha^i + \alpha^j \right) \right)$$

$$\leq \pi_b \sum_{i=0}^{T} \left( \frac{1}{k} \alpha^i + \sum_{s=1}^{k} P_{a,s} \left( \alpha^i + \frac{1}{T-i+1} \frac{1-\alpha^{T-i+1}}{1-\alpha} \right) \right)$$

$$\leq \pi_b \sum_{i=0}^{T} \left( \frac{1}{k} \alpha^i + \sum_{s=1}^{k} P_{a,s} \left( \alpha^i + \frac{1}{T-i+1} \frac{1}{1-\alpha} \right) \right)$$

$$\leq \pi_b \sum_{i=0}^{T} \left( \frac{1}{k} \alpha^i + \alpha^i + \frac{1}{T-i+1} \frac{1}{1-\alpha} \right)$$

$$\leq \pi_b \sum_{i=0}^{T} \left( \frac{1}{k} \alpha^i + \alpha^i + \frac{1}{T-i+1} \frac{1}{1-\alpha} \right)$$

$$\leq \pi_b \left( \left( 1 + \frac{1}{k} \right) \frac{1-\alpha^{T+1}}{1-\alpha} + \frac{\log(T+1)+1}{1-\alpha} \right)$$

$$\leq \pi_b \frac{2 + \frac{1}{k} + \log(T+1)}{1-\alpha}$$

$$\leq \frac{2 \log T}{1-\alpha}$$

$$= \frac{\log T}{\min_{i,j} P_{i,j}}$$

$$= O(\log T)$$

The last step follows from our assumption, completing the proof. $\qquad \square$

**Lemma B.7.** *Let $P$ be a stochastic matrix with all positive entries, and let $a, b$ be states. Assume that $\min_{i,j} P_{i,j}$ is positive and doesn't depend on $T$. Then,*

$$\sum_{s=1}^{k} \sum_{u=1}^{k} \left( \frac{1}{k} - P_{u,s} \right) \sum_{i=a}^{T} \frac{\pi_u}{T-i+1} \left( P^i \right)_{s,u} \left( (P^a)_{u,s} - \frac{1}{T-i+1} \sum_{j=0}^{T-i} \left( P^j \right)_{u,s} \right)$$

$$= (\log(T+1) - \log(a+1)) \sum_{s=1}^{k} \sum_{u=1}^{k} \pi_u^2 P_{u,s} \left( \pi_s - (P^a)_{u,s} \right) + O(1)$$

*Proof.* First notice that,

$$\sum_{s=1}^{k} \sum_{u=1}^{k} \left( \frac{1}{k} - P_{u,s} \right) \pi_u^2 \left( (P^a)_{u,s} - \pi_s \right) = \sum_{s=1}^{k} \sum_{u=1}^{k} \pi_u^2 \left( \frac{1}{k} \left( (P^a)_{u,s} - \pi_s \right) - P_{u,s} \left( (P^a)_{u,s} - \pi_s \right) \right)$$

$$= \sum_{u=1}^{k} \pi_u^2 \left( \sum_{s=1}^{k} \frac{1}{k} \left( (P^a)_{u,s} - \pi_s \right) - \sum_{s=1}^{k} P_{u,s} \left( (P^a)_{u,s} - \pi_s \right) \right)$$

$$= \sum_{u=1}^{k} \pi_u^2 \left( \frac{1}{k} (1-1) - \sum_{s=1}^{k} P_{u,s} \left( (P^a)_{u,s} - \pi_s \right) \right)$$

$$= \sum_{u=1}^{k} \pi_u^2 \sum_{s=1}^{k} P_{u,s} \left( \pi_s - (P^a)_{u,s} \right)$$

We will bound the distance between $\sum_{s=1}^{k} \sum_{u=1}^{k} \left( \frac{1}{k} - P_{u,s} \right) \pi_u^2 \left( (P^a)_{u,s} - \pi_s \right)$ and $\mathbb{E}_{x|P} \left[ \frac{\partial L_T}{\partial v_a} \right]$.
Define $\alpha = 1 - 2 \min_{i,j} P_{i,j}$ as in lemma B.5.

$$= \left| \sum_{s=1}^{k} \sum_{u=1}^{k} \left( \frac{1}{k} - P_{u,s} \right) \sum_{i=a}^{T} \frac{\pi_u}{T-i+1} \left( P^i \right)_{s,u} \left( \left( P^a \right)_{u,s} - \frac{1}{T-i+1} \sum_{j=0}^{T-i} \left( P^j \right)_{u,s} \right) \right.$$

$$\left. - \sum_{s=1}^{k} \sum_{u=1}^{k} \left( \frac{1}{k} - P_{u,s} \right) \sum_{i=a}^{T} \pi_u^2 \frac{1}{T-i+1} \left( \left( P^a \right)_{u,s} - \pi_s \right) \right|$$

$$= \left| \sum_{s=1}^{k} \sum_{u=1}^{k} \left( \frac{1}{k} - P_{u,s} \right) \sum_{i=a}^{T} \frac{\pi_u}{T-i+1} \left( \left( P^i \right)_{s,u} \left( \left( P^a \right)_{u,s} - \frac{1}{T-i+1} \sum_{j=0}^{T-i} \left( P^j \right)_{u,s} \right) - \pi_u \left( \left( P^a \right)_{u,s} - \pi_s \right) \right) \right|$$

$$\leq \sum_{s=1}^{k} \sum_{u=1}^{k} \sum_{i=a}^{T} \frac{\pi_u}{T-i+1} \left| \left( P^i \right)_{s,u} \left( \left( P^a \right)_{u,s} - \frac{1}{T-i+1} \sum_{j=0}^{T-i} \left( P^j \right)_{u,s} \right) - \pi_u \left( \left( P^a \right)_{u,s} - \pi_s \right) \right|$$

$$\leq \sum_{s=1}^{k} \sum_{u=1}^{k} \sum_{i=a}^{T} \frac{\pi_u}{T-i+1} \left| \left( P^i \right)_{s,u} \left( \pi_s - \frac{1}{T-i+1} \sum_{j=0}^{T-i} \left( P^j \right)_{u,s} \right) - \left( \left( P^a \right)_{u,s} - \pi_s \right) \left( \left( P^i \right)_{s,u} - \pi_u \right) \right|$$

$$\leq \sum_{s=1}^{k} \sum_{u=1}^{k} \sum_{i=a}^{T} \frac{\pi_u}{T-i+1} \left| \left( P^i \right)_{s,u} \frac{1}{T-i+1} \sum_{j=0}^{T-i} \left( \pi_s - \left( P^j \right)_{u,s} \right) - \left( \left( P^a \right)_{u,s} - \pi_s \right) \left( \left( P^i \right)_{s,u} - \pi_u \right) \right|$$

$$\leq \sum_{s=1}^{k} \sum_{u=1}^{k} \sum_{i=a}^{T} \frac{\pi_u}{T-i+1} \left( \left( P^i \right)_{s,u} \frac{1}{T-i+1} \sum_{j=0}^{T-i} \left| \pi_s - \left( P^j \right)_{u,s} \right| + \left( \left( P^a \right)_{u,s} - \pi_s \right) \left| \left( P^i \right)_{s,u} - \pi_u \right| \right)$$

$$\leq \sum_{s=1}^{k} \sum_{u=1}^{k} \sum_{i=a}^{T} \frac{\pi_u}{T-i+1} \left( \left( P^i \right)_{s,u} \frac{1}{T-i+1} \sum_{j=0}^{T-i} \alpha^j + \left( \left( P^a \right)_{u,s} - \pi_s \right) \alpha^i \right) \qquad \text{By lemma B.5}$$

$$= \sum_{s=1}^{k} \sum_{u=1}^{k} \sum_{i=a}^{T} \frac{\pi_u}{T-i+1} \left( \left( P^i \right)_{s,u} \frac{1}{T-i+1} \frac{1 - \alpha^{T-i+1}}{1-\alpha} + \left( \left( P^a \right)_{u,s} - \pi_s \right) \alpha^i \right)$$

$$\leq \sum_{s=1}^{k} \sum_{u=1}^{k} \sum_{i=a}^{T} \frac{\pi_u}{T-i+1} \left( \frac{1}{T-i+1} \frac{1}{1-\alpha} + \left( \left( P^a \right)_{u,s} - \pi_s \right) \alpha^i \right)$$

$$\leq \sum_{i=a}^{T} \frac{1}{T-i+1} \left( \sum_{s=1}^{k} \frac{1}{T-i+1} \frac{1}{1-\alpha} + \sum_{s=1}^{k} \left( \sum_{u=1}^{k} \pi_u \left( P^a \right)_{u,s} - \pi_s \right) \alpha^i \right)$$

$$\leq \sum_{i=a}^{T} \frac{1}{T-i+1} \left( \sum_{s=1}^{k} \frac{1}{T-i+1} \frac{1}{1-\alpha} + \sum_{s=1}^{k} \left( \pi_s - \pi_s \right) \alpha^i \right)$$

$$\leq \sum_{i=a}^{T} \frac{1}{T-i+1} \left( \frac{k}{T-i+1} \frac{1}{1-\alpha} \right)$$

$$\leq \sum_{i=a}^{T} \frac{1}{(T-i+1)^2} \frac{k}{1-\alpha}$$

$$\leq \frac{2k}{1-\alpha}$$

$$= \frac{k}{\min_{i,j} P_{i,j}}$$

$$= O(1)$$

The last step follows from our assumption, and the fact that $k$ does not depend on $T$. $\qquad\qquad \square$

