# OpenReview forum: "The Evolution of Statistical Induction Heads: In-Context Learning Markov Chains"
_NeurIPS.cc/2024/Conference — NeurIPS 2024 poster_

### Official Review · Reviewer_1EuX · 2024-07-11

**Soundness:** 3
**Presentation:** 3
**Contribution:** 2
**Rating:** 6
**Confidence:** 3

**Summary:**

This work introduces a simple “in-context learning” Markov chain modelling task and studies how transformer models tackle it. The task consists of inferring the transition probability matrix of the Markov chain, which is sampled from a prior Dirichlet distribution. The authors consider the task to pertain to “in-context learning” because each data point (that is, each sequence in the dataset) is sampled from a different transition probability matrix. The prior over the matrices is however global and thus shared by both training and test datasets. More in detail, the authors focus on Markov chains with only two states, and sample each sequence from the stationary distribution of the process.

Given this dataset, the authors investigate the training dynamics of a two-layer attention-only transformer model both empirically and theoretically.  The theoretical investigation involves the introduction of a simplified model, which is defined to mimic the main features of the transformer model.  The authors report three different training stages, during which the model very quickly changes from predicting random transitions, to predicting samples similar to the stationary distribution of the chain, and lastly, predicting transition from the inferred transition probability matrix.

The authors further connect their findings with previous work.

**Strengths:**

Deep learning architectures are remarkably complex systems and designing both tractable models and modelling tasks that mimic the behaviour of these systems, prior, during or after the training process, is an important approach to unveil their inner workings. This paper attempts to carry one such analysis, by introducing a simple “in-context learning” Markov chain modelling task and studying how a two-layer attention-only transformer model and a simplification thereof solve it, thereby adding another interesting contribution to our understanding of transformer models.

One strength of the paper is that the authors first demonstrate that both their models can indeed find the optimal solution of their Markov chain task, and then empirically verify this, which gives soundness to their claims.  In particular the results in Figure 3, which demonstrate the similarity in behaviour of both models, are very compelling (see however the questions below). A second strength is that the authors identify different phenomena within their setting which have also been observed before, like multiple distinct stages of training, the latter of which is also connected to induction head formation, or the order in which the layers in the network are learned. Finally, the authors demonstrate that some of their findings are also present in a second-order Markov chain modelling task.

Putting aside some minor details with some notation and content organisation (see below), the paper is overall well written.

**Weaknesses:**

My first issue with the paper is:

- how much does the proposed learning task really concern “in-context learning”? especially in the context of large language models, which the authors use as motivation.

As framed by e.g. Xie et al. (2022) who focus on LLMs, "in-context learning" deals with sequences which have low probability under the training distribution. The authors do not really elaborate in their understanding or interpretation of "in-context learning", nor do they explain how it relates to "in-context learning" in the setting of language models. In short, I believe the paper contributes more to our understanding of transformers than to our understanding of in-context learning.

A second major issue is that the authors do not explain their reasoning behind nor the limitations of their dataset/task definition.

- First, the authors consider Markov chains with only two states and do not comment on why they restricted their study to that case nor how their findings extend to Markov chains with more states, if at all.
- Second, the authors choose the initial distribution of the chain to be its stationary distribution but, again, did not explain why. Note that a stationary Markov chain corresponds to a very special case, and it's not very clear how the observations made by the authors extend beyond it.  Indeed, choosing a Markov chain with more states, initialised from a distribution far from stationarity, can generate long sequences which are “out-of-equilibrium”. One can't but wonder whether one would still find multiple training stages in this case. See e.g. question 8 below.

Despite its merits, I think the issues above require some revision, or more detailed explanation, before the paper can be published.

*Other comments*:

The comment in line 205 “We observe that training a 1-layer transformer fails to undergo a phase transition or converge to the right solution”, could be better highlighted as evidence for the emergence of induction heads. It feels somewhat buried in between other comments and statements. Also there’s a typo in line 239 and another in line 280.

*References:*
- An explanation of in-context learning as implicit Bayesian inference. Xie et al. (2022)

**Questions:**

1. Why did you consider Markov chains with two states only?
2. Can you please elaborate on your comment in line 145, page 4: “the stationary distribution of this random Markov chain does not admit a simple analytical characterization when there is a finite number of state”?
3. Why did you choose the stationary distribution as the initial distribution?
4. Is the ground-truth unigram strategy then computed by estimating the stationary distribution with the count histogram? Is this the one used to compute the KL distance in Figure 3?
5. Similarly, do you use the bigram strategy of section 2.1 to compute the corresponding KL distance in Figure 3? Or do you use the instance-dependent ground-truth transition probability matrix instead?
6. Is the difference in KL between the unigram and bigram strategy due to finite sampling?
7. Why does the bigram strategy have a smaller loss? is it because of the inductive bias of the model?
8. In the setting with an initial condition far from the stationary distribution, would the model first fit the step-dependent marginal distribution of the chain (a unigram strategy) and later fit the bigram strategy of section 2.1? Isn't it simpler to fit the latter first (as opposite to the stationary case)?
9. Is the 2-layer transformer model initialised to the matrices in Appendix C.1?
10. The sequence length is first labelled by $t$ (e.g. after eq. 1) and then by $T$ (e.g. before eq. 4). Is this change of notation intentional? Or am I misunderstanding something?

**Limitations:**

Yes, they authors did address (some of) the limitations of their method.

---

> ### Author Rebuttal · Authors · 2024-08-07
>
> Thank you for critically reading our work and helping us improve its presentation. We greatly value your in-depth review!
>
> We will first discuss your most significant concerns with our work:
>
> ## In-Context Learning
>
> You're right that we don't spend much time in our paper discussing what "in-context learning" means, and indeed, there isn't a single consensus definition! The term "in-context learning" was (we believe) introduced in the GPT-3 paper ([Henighan et al., 2020]) to refer specifically to the phenomenon where LLMs have improved few-shot performance with increasingly many examples in their context. Xie et al., meanwhile, does indeed explore scenarios where there is a distribution shift between training and inference; however, in the broader literature, ICL is not consistently tied to distribution shifts (see, e.g. [Garg et al., 2022] for an influential study of ICL that does not require a distribution shift in its definition). Meanwhile, [Elhage et al., 2021], which was roughly concurrent with Xie et al., and the follow-up work [Olsson et al, 2022] -- use ICL very broadly to refer to LLMs being better at predicting tokens that appear later in a sequence. The way we use the term is somewhat narrower than this; as we state in the Introduction: we think of ICL as when language models are "incorporating patterns from their context into their predictions." In other words, it is when models *learn* from patterns in their context; few-shot learning being a special case of this.
>
> The task we focus on, ICL-MC, is an in-context version of a classic learning task, Markov chain inference. Each test-time sequence has *zero* probability of appearing during training, because of the continuous nature of the distribution over chains. During pre-training, the transformer learns to perform the learning task in-context. We don't only focus on the case where there is a distribution shift between train and test -- as discussed above, we think ICL is interesting even in the absence of distribution shift. (Though we do explore distribution shifts in the paper: see Figures 4 and 11.)
>
> We would be happy to elaborate on our interpretation of ICL in the camera-ready version of the paper.
>
> ## Answers to questions
> 1\. This choice was made to facilitate a theoretical analysis of the observed phenomena. In particular, results such as Lemma C.2 become feasible only when the number of states is 2, or other simplifying assumptions are made. However, in the experiments, we have verified that our insights transfer to Markov Chains with a larger number of states (see, for example, Figure 7 in the Appendix). We will add a discussion of Markov chains with more states, including a reference to figure 7.
>
> Based on your concerns, we have decided to add back in a proof we decided not to include, which means that **by making only small changes to the submitted version, we are able to extend our main results to any number of states**. This requires using a distribution like that in experiment on the left of figure 4. Due to lack of space, we defer the details of this to a comment.
>
> 2\. Thank you for pointing out this line as being out of place. The line (edited for clarity) will be moved to the discussion of limitations. It intended to communicate the difficulties with the distribution of the stationary distribution for $k>2$. Due to lack of space, we defer further elaboration to a comment.
>
> 3\. We chose this to make the setting closer to the next token prediction seen in practice. The marginal distributions of the tokens generated from a markov chain approach the stationary distribution exponentially fast no matter the initial distribution.
>
> We have ran experiments for other somewhat natural starting distributions (such as uniform) and found no meaningful differences.
>
> 4\. Yes. We will add the following mathematical definition: the unigram strategy's probability of the token at position $t+1$ being $j$ is $\frac{1}{T}\sum_{i=0}^T \mathbb{1}[x_i=j].$
>
> 5\. The bigram strategy is computed by counting the frequency of pairs of states in the context of each sequence. This corresponds to the formula in line 157.
>
> The graphs on the left in figure 3 show the KL-div loss using the instance-dependent ground-truth transition probability matrix. Because the context size is long (100 tokens), and we average over 1024 test samples, these two measures are almost identical (as we would expect).
>
> 6\. No, the KL div between the unigram and bigram strategies is not due to finite sampling. The unigram strategy is a flawed strategy (it ignores the information we get from the relative order of the tokens), for almost all markov chains it will get higher expected loss than the bigram strategy, even for small context sizes.
>
> 7\. The bigram strategy is a near optimal strategy (it approaches the optimal strategy exponentially fast as the context length grows). This is true regardless of the model used.
>
> 8\. We are not entirely sure how to interpret this question and would appreciate clarification.
>
> Sampling the first token by the stationary or uniform distribution does not meaningfully affect any empirical observations. The marginal distributions of the tokens after the first will always approach the stationary distribution exponentially fast.
>
> Empirically, the model starts by representing the unigram strategy and later the bigram strategy. In both cases it is not obvious that the model needs to fit the unigram strategy at any point (after all, the unigram strategy performs worse than the bigram strategy), and so it must be  only due to the inductive biases of the model and optimizer that this happens.
>
> 9\. No. The transformers were initialized with the default Pytorch initializations (Gaussian with mean 0), we will clarify this in section B.
>
> 10\. This is a typo.
> ***
> We hope we have managed to address your concerns and would be grateful if you adjusted your score accordingly.

---

> > ### Comment · Reviewer_1EuX · 2024-08-11
> >
> > I appreciate the authors' detailed response. Before I update my score, I have a couple more questions/comments, if possible. I apologize in advance for these additional inquiries; my intention is simply to gain a better understanding of some of the claims made in the paper. I hope these questions also help the authors present their results more clearly.
> >
> > @3: When you write:
> >
> > *"The marginal distributions of the tokens generated from a markov chain approach the stationary distribution exponentially fast no matter the initial distribution"*
> >
> > do you mean for $k=2$ states?
> >
> > Because one can construct transition probability matrices which yield Markov chain that exhibit slow convergence to stationarity. A random walk with periodic boundary conditions is an example, where the mixing times are of the order of $k^2$, for $k$ the number of states. Or am I missing sth?
> >
> > @6: Could you please remind me how you compute the KL divergence in your experiments? Meaning, you compute the KL wrt. what?
> >
> > @8: Let's assume that we are dealing with a Markov chain that exhibits slow convergence to stationarity. In such a case, the marginal unigram distribution changes at every step. Let's also assume one train your model on the *out-of-equilibrium* sequences sampled from said Markov chain. Do you still think the model will first find such a unigram strategy? I hope this rephrasing makes my question clearer.

---

> > > ### Author Response · Authors · 2024-08-12
> > > **Response to Reviewer Comment**
> > >
> > > @3: To clarify, in our setting $T$ is taken to be large, while $k$ is effectively a constant. This is similar to natural language settings, where the number of letters or tokens is some constant, but the context can be arbitrarily long. When we wrote that the  "marginal distributions... approach the stationary distribution exponentially fast", we were referring to Lemma C.4, which states that the distribution of the token n tokens after the current has distance to the stationary distributed bounded by $\alpha^n$, for some $\alpha<1$ depending on the specific chain.
> > >
> > > @6: In our experiments, we used KL divergence to measure the difference between the probabilities predicted by the model and other probability distributions. For test loss, this other distribution was the appropriate rows of the transition matrices used to generate the test examples.
> > >
> > > Formally, let $f(x_{1:T-1})$ be the softmax distribution of the transformer's output, given the input sequence $x_{1:T-1}$. In our standard setting, we measured
> > > $$d_{KL}(\mathcal{P}\_{x_{T-1}} || f(x_{1:T-1}))$$
> > > where $\mathcal{P}\_{x_{T-1}}$ is the true distribution of the next state $x_T$ given the previous state, under the true Markov chain $\mathcal{P}$. Note that $\mathcal{P}$ varies from sequence to sequence (it is drawn from a prior over transition matrices) and is not directly observable by the learner—this is what needs to be learned in-context.
> > >
> > > For measuring how close the model was to various strategies, we computed the predicted probabilities given by said strategies, and used those as the base distribution. Note that the  output of the bigrams strategy (which is Bayes-optimal for our base setting) is different from the aforenentioned ground-truth $\mathcal{P}\_{x_{T-1}})$. Instead, as described in Section 2.1, it is a Bayesian posterior distribution of the next state given the observed sequence, with the prior determined by the prior distribution of transition matrices. Formally:
> > > $$ \mathbb{E}[\mathcal{P}\_{x_{T-1}} | x_{1:T-1}] $$
> > > where the expectation is taken over the draw of Markov chain transition matrix.
> > >
> > > @8: Setting aside the mixing time issue discussed in @3, your question seems to suppose that all the training sequences are drawn from a single Markov chain, whereas in our work there is a new chain for each sequence. Do you have a more precise formulation of your question in mind? Regardless, we agree it could be interesting to explore what happens when the number of states is comparable to the sequence length, and the prior distribution over chains is crafted to favor chains that mix slowly, though this is beyond the scope of this work.

---

> > > > ### Comment · Reviewer_1EuX · 2024-08-12
> > > >
> > > > I thank the authors again for their responses. I will update my score.

---

> ### Author Response · Authors · 2024-08-07
> **New Lemma for $k>2$**
>
> We can prove a version of Lemma 3.1 for any number of states (k) as long as we consider a different distribution for transition matrices: specifically, a mixture of the distribution where the unigram strategy is optimal, and the distribution where the unigram strategy is as bad as guessing randomly. The proof only requires different versions of lemmas C.1 and C.2, and actually adds more intuition to our stage-wise learning story. Specifically, in the first phase of learning, the contribution to the gradient from the unigram-optimal distribution dominates, but in the second phase, the other component (from the distribution where unigrams are useless) is dominant. We would like to add this variation of Lemma 3.1 to our camera-ready version.

---

> ### Author Response · Authors · 2024-08-07
> **Stationary Distribution Results**
>
> For two states, the transition matrix can be represented as $\mathcal{P}=\begin{pmatrix}1-\alpha&\alpha\\\beta&1-\beta\end{pmatrix}$, where $\alpha,\beta$ are iid random variables from the uniform distribution on $[0,1]$. The stationary distribution is $\pi = \begin{pmatrix} \frac{\beta}{\alpha + \beta}, & \frac{\alpha}{\alpha + \beta} \end{pmatrix}$, which is feasible to analyze. While there exist results such as [Chafai et al](https://arxiv.org/abs/0808.1502) that characterize the stationary distribution in the limit as the number of states approaches infinity, we do not believe analogous results exist for any constant number of states greater than two.

---

### Official Review · Reviewer_aVXq · 2024-07-12

**Soundness:** 2
**Presentation:** 2
**Contribution:** 2
**Rating:** 5
**Confidence:** 4

**Summary:**

The paper studies in-context learning with transformer models in a simple Markov Chain sequence modelling task. The authors empirically show the formation of statistical induction heads which correctly compute the posterior probabilities given bigram statistics. Moreover, they observe that during training the model undergoes phase transitions where the complexity of the n-gram model increases. They also propose a simplified theoretical model of a two-layer transformer to analyse these phenomena.

**Strengths:**

1. The paper addresses a relevant topic within a simplified setting, facilitating the interpretability of transformer model solutions and an understanding of their training dynamics.
1. Utilising Markov chains is an effective approach to studying sequence-to-sequence modelling.
1. The study seeks to balance empirical investigation and theoretical analysis.
1. The observation that models may progress from simple solutions, like unigrams, to more complex structures during training is interesting.
1. It is relevant that transformers can learn an algorithm to estimate the transition matrix in context by gradient descent.

**Weaknesses:**

The major weakness of this work is that the abstract and introduction suggest a primary focus on analysing and uncovering the mechanisms behind the simplicity bias and the phase transition from simple unigram solutions to more complex ones. However, upon reading the main body of the paper, it seems that the experiments mainly reveal the existence of such biases and transitions, while the theoretical section addresses a different phenomenon: how the two layers are learned in different training phases. The plateau behaviour is particularly intriguing, such as the model learning unigrams, bigrams, trigrams, and so on, but their theory does not seem to describe or explain this observation.

1. **Unigram strategy:** Can the authors clarify this statement "*the stationary distribution of this random Markov chain does not admit a simple analytical characterization when there is a finite number of states*". In this context, what does "*this Markov chain*" refer to?

1. **Simplified transformer:** the proposed simplified transformer seems to be able to capture all the phenomenology of a real transformer. Nevertheless, it is not clear to me if the proposed model is a good proxy for an attention-only transformer or not. In particular:
	1. The first attention layer does not use the interaction between tokens but represents the attention through a learnable matrix that could in principle learn the same structure. Could the authors elaborate on this choice?
	1. The second attention instead captures the interaction between tokens but between the input and the output of the first layer which is unusual. I understand that this is based on the construction of the real transformer where the output of the first attention is copied in the second block of the embedding and therefore can interact with the input but it still means that the simplified version doesn't need to learn this mechanism. Could the author elaborate on this choice?
	1. The model seems to be composed of a non-linear attention for the first layer (softmax is present) and linear attention for the second. Could the authors elaborate on this particular choice? Is it a way to simplify the analysis while maintaining the properties of the softmax where needed?
1. **Data generation:** in section 2, it is explained how the transition matrices are generated according to a Dirichlet distribution, nevertheless in the theory and some of the experiments there is the additional requirements for the matrix to be doubly stochastic which is not mentioned when the setup is described. Could the authors clarify this point and highlight the importance of doubly stochastic transition matrices?
1. **Proof of Proposition 2.2:** the proof of this proposition appears disorganised and not easy to follow. In particular:
	1. Setting the internal dimension d=3k seems fundamental to ensure that the model has the correct number of dimensions to copy the tokens from one layer to the next. I do not necessarily have a problem with this choice but it would be useful to see it discussed in the main text.
	2. Is the definition of $v^{(1)}$ correct? if I am not mistaken it appears that the matrix is of dimensions $3k \times 3k$ whereas by the main text, it should be of dimensions $t \times 3k$ could the author clarify this together with the definition of $\delta_2$ and $1_k$.
	3. There are multiple quantities used in the text such as $e_{x_i}$ or $e_i$ or $e_{{i-1},j}$ could the authors clarify their meaning.
	4. In the expression $\text{softmax}(\text{mask}(A))_{i,j} \approx \mathbb{1}[j=i-1]$ shouldn't it be  $[i=j-1]$ ? Why is the indicator function comparing the indexes instead of the values $x_i$ ? Moreover, could the authors clarify why it is only the indicator function and not $\frac{1}{\text{count}([x_i=x_j-1])}$ given the softmax?
	5. In the expression of $\text{Attn}_2(e)$ why is the summation from $h=1$ to $h=3k$ if h is the index of the element in the sequence? Furthermore a new index $g$ appears which is not used anywhere.
1. **Experiments in Figure 8:** are the experiments in Figure 8 using d=3k ? do the authors observe that the parameters converge to the construction given in Proposition 2.2 ?
1. **Proof of Proposition 2.3:** The proof for the unigram construction seems to be missing.
1. **Unigram and bigram constructions:** For both the two-layer transformer and the simplified model it is possible to show constructions for the unigram and bigram models. Nevertheless, besides giving the weights for such constructions the authors do not discuss the relationship between them and the objective functions. Are they stationary points of the dynamics? Does this help explain the plateau observed in the experiments? is the unigram a saddle point?
1. **Minimal model:** In line 241 the authors state that "*the minimal model converges to the bigram solution spending however significantly less time at the unigram solution*".  By looking at Figure 3 I understand how the minimal model reaches the same Kl value in almost half the time but it also has fewer parameters and some of the structure is already enforced by construction (for example the fact that the second attention is between input and output of the first layer). Could the authors explain why the comparison still makes sense?
1. **Varying the data distribution:** I find this part unclear. In line 231 the authors state that they define distributions  "*we define distributions over Markov chains that are in between the distribution where unigrams is Bayes optimal, and the distribution where unigrams is as good as uniform.*" can the authors provide a mathematical definition of such distribution?   Are doubly stochastic transition matrices used to create Markov chains with uniform stationary distribution such that the latter would be the only unigram possible?
1. **Two phases of learning in Lemma 3.1:** The model is capable of reproducing the effect of learning the second layer first, nevertheless I am not convinced that this is a property of the model of the simplified transformer but rather a consequence of the initialization and step size. Could the authors clarify this point?

**Questions:**

See above.

**Limitations:**

- Simplified transformer architecture
- Synthetic task only, not clear if similar phenomena appear in larger models trained on natural language data.

---

> ### Author Rebuttal · Authors · 2024-08-07
>
> Thank you for the in-depth review and feedback!
>
> You point out that there is some misalignment between our experimental story and our proofs, and you see that as our paper's major weakness. We agree that there is misalignment, due to the difficulty of analyzing SGD on multi-layer Transformers. However, we believe our analysis is very relevant to the experimental observations about simplicity bias and phase transitions. In particular, Lemma 3.1 exhibits simplicity bias by showing that the parameters after the first phase compute a solution that is a weighted version of the unigram solution, and the second phase computes a mixture of unigram and bigram. Also, the magnitude of the signal for the second phase is a factor $T$ lower than the first step, indicating that the second phase requires more time for "signal accumulation". We agree that this does not explain the sharpness of the phase transition, but it does provide theoretical intuition for the existence of multiple phases. We are happy to add these aspects of the story to the discussion following Lemma 3.1 in the text; for these reasons, we don't think the extent of misalignment is a serious weakness in the paper.
>
> We address your various minor concerns below:
>
> 1\. Thank you for pointing out this line as being out of place. A version of the line (edited for clarity) will be moved to our discussion of limitations. The line intended to communicate the difficulties with understanding the stationary distribution when $k>2$.
>
> 2.1 and 2.2. The goal of the minimal model is to capture the dynamics of the full transformer, while being amenable to theoretical analysis. Essentially all analyses of multilayer transformers have relied on simplifying out many parts of the transformer. While simplifying the transformer we carefully checked that the dynamics we wished to understand were preserved as much as possible. Due to space limitations, we have deferred a more in depth explanation of the choices in the minimal to a comment.
>
> 2.3. This is correct. Unfortunately, the analysis does not seem to be feasible with softmax in both layers.
>
> 3 and 9. Generating each row of the transition matrix iid from a flat Dirichlet distribution is equivalent to uniform distribution over all transition matrices. We chose to focus primarily on this distribution because it is the most natural.
>
> The only mention of doubly stochastic matrices in the theory (lemma C.3) was mistakenly left in after the surrounding context was removed.
>
> Due to space constraints, we defer definitions of the other distributions to a comment.
>
> 4\. Thanks for your careful proofreading. Due to space constraints we defer the specific changes and improvements made in response to a comment.
>
> 5\. The trained transformers used an internal dimension of 16. The specific parameters after training always vary randomly due to initialization and training samples. Figure 8 shows that the transformers are implementing the same algorithm as our construction, with each layer performing the same functions.
>
> 6\. Thanks for pointing out this oversight, we have added the full proof in (it's rather short and simple).
>
> 7\. That's an interesting question! Unfortunately, the details of the loss landscape for even simplified transformers are hard to precisely characterize with current theoretical tools. Among other difficulties, a barrier to transformer optimization results is analyzing cases where the input to softmax isn't close to zero, or one hot.
>
> Our intuition is that the unigram is not exactly a saddle point: during the plateau period, there is very slow continual improvement in the loss. We suspect the gradient at the unigram solution is simply small relative to the change needed to move towards the bigram solution.
>
> 8\. Line 241 is a holdover from a previous version, and no longer applies to the experiments in the paper, we apologize that it made into this version and will remove it.
>
> On the left in figure 3, both the minimal model and transformer model reach their minimum loss after seeing around 80,000 training sequences. We do believe though that one shouldn't put too much weight into the fact that these took the same time to train.
>
> We believe this comparison still makes sense. The minimal model is designed to be able to express everything the transformer expresses when being trained on the task. Experimentally, it goes through the same phases. In the right most graphs in figure 3, both models quickly learn to use the unigram strategy, before eventually adopting the bigram strategy. Since this task does not require the full expressive power of a transformer, it is not too surprising that a model with many fewer parameters is sufficient to gain insight into the training dynamics.
>
> 10\. The most compelling piece of evidence that the minimal model actually goes through these phases are the experimental results shown in figures 3 and 4. Experimentally this also works for standard small Gaussian initializations. In the conclusion, we will add a mention of the limitation of this being a two step analysis. Unfortunately, existing tools are insufficient to analyze dynamics involving softmaxes (specifically when the inputs to the softmax are neither small nor dominated by a single index). For lemma 3.1 we were forced to limit our analysis to two steps. However, as we mention on lines 579-580, the uniform component of $W$ does not contribute to the gradient of $v$, so even if we initialized $W$ to any uniform non-zero initialization, $v$ would still not change in the first step. We chose the specific step sizes to allow the theory to show how the model can go through the unigram and bigram phases, which happens in practice over the course of many small steps.
> ***
> We hope we have managed to address your concerns. If you think we have adequately done so, we would be grateful if you adjusted your score accordingly. Thank you!

---

> ### Author Response · Authors · 2024-08-07
> **Minimal Model Design**
>
> To create our minimal model, we started from a two layer attention-only disentangled (see [Elhage et al](https://transformer-circuits.pub/2021/framework/index.html)) transformer (using relative positional embeddings) and iteratively simplified parts that empirically did not affect the training dynamics. In our construction, the first layer only attends to positional embeddings, and the second layer ignores positional embeddings, so we set the first layer key matrix and the positional embeddings from the second layer to zero. Then we set the value matrices and query matrices to the identity in both layers. In the experiments, all optimizations such as layer norms and weight decay were removed, and the optimizer used was SGD. With all of these changes, the overall training dynamics were not changed much at all, depending on hyper parameters they could speed up training, but the same loss curve and phases were observed. To make analysis of gradient descent on the minimal model feasible, the softmax on the second layer had to be removed, which did make the phase transition less sharp.

---

> ### Author Response · Authors · 2024-08-07
> **Doubly stochastic and 'unigram' distributions**
>
> Doubly stochastic transition matrices are those for which the stationary distribution is the uniform vector, or those for which the unigram strategy and the uniform strategy get the same loss. If we want to observe the inductive biases of the model when there is no signal encouraging the unigram strategy, then sampling from doubly stochastic transition matrices is natural.
>
> Finally, we also considered the distribution where each row in the transition matrix is the same, resulting in the unigram and bigram strategies having the same loss.
>
> The mathematical definition of the mixed distribution in the graph on the left of figure 4 is as follows: with 75% chance, choose a uniformly random doubly stochastic transition matrix, otherwise make each row of the transition matrix the same vector, chosen from a flat Dirichlet distribution.

---

> ### Author Response · Authors · 2024-08-07
> **Question 4**
>
> 4.1. We use a dimension of $3k$ to have a simple and intuitive construction. In practice, models can learn with a far smaller internal dimension. We can add discussion of this to the main text.
>
> 4.2. We apologize for the confusing notation with regard to the matrix definitions, it will be improved. $v^{(1)}$ is of dimension $t\times d$, where $d=3k$. We will make sure that the dimensions of each submatrix used to define $v^{(1)}$ and all other matrices are all specified in the notation.
>
> 4.3. $x_i$ refers to the token at position $i$. $e$ is used to represent the $1$-hot embeddings, $e_i$ is a vector that is $0$s besides at position $i$ where it is $1$. Similarly, $e_{x_i}$ is all zeros except at position $x_i$ where it has a $1$. $e_{i-1,j}$ is the $j$th index of the vector $e_{i-1}$, hence $e_{i-1,j}=\mathbb{1}[i-1=j]$. We will do an extra pass over all of this notation to clarify and improve it.
>
> 4.4. and 4.5. We thank you for pointing out these typos. It should say:
> $$\text{softmax}(\text{mask}(A))\_{i,j}\approx \frac{\mathbb{1}[x_{j-1}=i]}{\sum_{h=1}^i \mathbb{1}[x_{h-1}=i]}$$
>
> Fixing these typos, we get the result
> $$Attn_2(e)\_{i,j+2k} =\frac{\sum_{h=1}^{k}\mathbb{1}[x_{h-1}=x_i]\mathbb{1}[x_h=j]}{\sum_{g=1}^i\mathbb{1}[x_{g-1}=x_i]}$$
> Which is more correct, since this is the bigram probabilities, instead of just the bigram statistics. That is, this is the empirical approximation of $P_{x_i, j}$.

---

> > ### Author Response · Authors · 2024-08-13
> >
> > Following up to see what you think about our response to your review. Let us know if you have any further questions or need any further clarifications?

---

> > > ### Comment · Reviewer_aVXq · 2024-08-13
> > >
> > > After considering your clarifications, I acknowledge that despite the multiple typos and minor concerns that affect the readability of the paper and the communication of the results, your work contains some insightful observations. These contributions are valuable and, therefore, I have decided to increase my score.

---

### Official Review · Reviewer_Kgpp · 2024-07-17

**Soundness:** 3
**Presentation:** 3
**Contribution:** 2
**Rating:** 5
**Confidence:** 2

**Summary:**

This paper introduces a task to investigate how in-context learning capabilities are learnt by transformer models. They show that models trained on this task go through a phase transition from which they start by modeling unigram to then acting as a bigram model. The authors further extend their work to the case of n=3 and show similar behavior.

**Strengths:**

1. Nice presentation, with clear figures exhibiting the key behavior in question (sudden "emergence" of the correct behavior.
2. Good attempt to form a theoretical/mathematical foundation, which extends to the appendix.
3. Attempt to understand ICL LLMs using a good toy task.

**Weaknesses:**

1. Questionable impact since this topic has been explored heavily in the past years, with other similar tasks and toy models existing, showing similar results.
2. Questionable how well the toy setting can actually transfer to real-world ICL, albeit interesting.

**Questions:**

What is the key differentiating contribution of the work in comparison with the myriad of other works in this space?

What is the conclusion of the work that you believe is transferable to real-world ICL setting. What n-gram statistics would that follow?

**Limitations:**

It would potentially be useful to also discuss the limitations of the toy task to explain the full-fledged LLM ICL setting. A lot has been said in the paper about conclusions made, and I understand that this the same with any work employing a toy task, but an attempt for making more direct comparisons between real vs. toy setting may enhance the value of the work.

---

> ### Author Rebuttal · Authors · 2024-08-07
>
> Thank you for your review and questions. Here we address the main weaknesses and questions raised by you.
>
> ## Impact and related works
> >Questionable impact since this topic has been explored heavily in the past years, with other similar tasks and toy models existing, showing similar results... What is the key differentiating contribution of the work in comparison with the myriad of other works in this space?
>
> As we discuss in our related work section, there are indeed prior works which introduced synthetic settings for exploring ICL:
> - [Garg et al., 2022](https://arxiv.org/abs/2208.01066) (and follow-ups) train models to learn linear functions (and other simple function classes) in a few-shot setting.
> - [Xie et al. 2022](https://arxiv.org/pdf/2111.02080) describe a few-shot learning setting in which each document consists of examples generated by a hidden Markov model.
> - Finally, most related to our work is [Bietti et al. 2023](https://arxiv.org/abs/2306.00802), in which all sequences are generated from a single Markov chain (which doesn't need to be learned in-context), but certain 'trigger' tokens are automatically followed by sequence-specific tokens which need to be learned in-context.
>
> We believe that our task, in-context learning of Markov chains (and the higher-order generalizations thereof), is a valuable contribution in the context of this rich literature. (No single synthetic task is going to capture all of the scientifically interesting aspects of in-context learning.) In particular:
> - It is very natural and simple to describe.
> - It elicits the formation of *induction heads* in networks trained on the task. (It is a particularly natural setting for studying their emergence.)
> - There are multiple strategies of various levels of sophistication which can be used to solve the task to varying degreees of success (unigram, bigram, etc.), and which networks pass through in stages during training.
>
> Moreover, the results we obtained by studying this task experimentally and theoretically are novel. Our key findings -- the formation of statistical induction heads, the stage-wise phase transitions between in-context solutions (unigram, bigram, trigram, ...) on the way to success, the arguments that simplicity bias may delay the formation of the correct in-context solution, and finding that the second layer of the model is learned before the first layer, rather than the other way around -- are all original to our work.
>
> Given that you say there are lots of very similar works, **we suspect you may be mistakenly assuming that some concurrent works were actually prior works.** Indeed, at the same time we released our preprint, there were several simultaneous papers (all with pre-prints posted within a single month) that shared aspects of our setup and/or scientific focus. Please see the "Concurrent Works" paragraph in our Related Work section for a discussion of these papers and their relation to our work:
>  [Akyürek et al., 2024](https://arxiv.org/abs/2211.15661), [Hoogland et al., 2024](https://arxiv.org/abs/2402.02364), [Makkuva et al., 2024a](https://arxiv.org/abs/2402.04161), and [Nichani et al., 2024](https://arxiv.org/abs/2402.14735). There have also been subsequent works in this space, including [Rajaraman et al., 2024a](https://arxiv.org/abs/2407.17686), [Rajaraman et al., 2024b](https://arxiv.org/abs/2404.08335) and [Makuva et al., 2024b](https://arxiv.org/abs/2406.03072).
>
> We want to make sure that you are following the scientific best practice and judging our work's significance in relation to *prior works*, not concurrent works. If you mistakenly thought the above concurrent works were prior, we understand! There has indeed been a burst of activity on this topic recently. In that case, we hope you will reassess our work in its proper context, and adjust your score accordingly. If you were referring only to actual prior works, we hope you find the above contextualization of our work convincing and reassess accordingly; if not, we would appreciate some elaboration on the works you are referring to.
>
>
> ## Real-world relevance
> > What is the conclusion of the work that you believe is transferable to real-world ICL setting. What n-gram statistics would that follow?
>
> N-gram statistics are useful for predicting real-world natural language text. Indeed, historically, language models were often simply n-gram predictors! It is also useful for models to learn in-context bigram (and more generally, n-gram) statistics. For instance, the writing style of a particular document is connected to the n-gram statistics of that document.
>
> The most concrete connection the toy MC-ICL setting has to real world ICL settings is induction heads, which have been shown to have importance to in-context learning in LLMs in past work ([Olsson et al, 2022](https://transformer-circuits.pub/2022/in-context-learning-and-induction-heads/index.html)). In the MC-ICL setting, the emergence of induction heads corresponds to the phase transition in loss, just like the phase transition found in the ICL ability of transformers hypothesized to be caused by induction heads emerging.
>
> Additionally, compared to other ICL tasks studied in the past (such as linear of logistic regression) we believe that n-grams are a better model for natural language specifically.

---

> > ### Author Response · Authors · 2024-08-12
> > **Link correction**
> >
> > A minor fix to our rebuttal: the reference to Akyürek et al., 2024 should point to [this paper](https://arxiv.org/abs/2401.12973).

---

> ### Comment · Reviewer_Kgpp · 2024-08-09
>
> Thank you for answering my questions. I admit that some, but definitely not all of the papers in mind may be concurrent works. I have read the other reviewer's comments, and in general it seems like while there is an agreement that this is a well-done piece of scientific literature, the impact is questionable, as the toy-setting field particularly when it pertains to ICL is definitely crowded. Finding new angles to attack the problem is a worthwhile pursuit, but at times may distract from perhaps other more important and less-studied issues.
>
> As I see there are better reviewers than me to assess this work, so I will raise my score from a 4 to a 5 while keeping my confidence low, and leave it up to them to reach a consensus.

---

### Official Review · Reviewer_oy83 · 2024-07-19

**Soundness:** 3
**Presentation:** 3
**Contribution:** 2
**Rating:** 6
**Confidence:** 3

**Summary:**

In the paper, the authors investigate the phenomenon of in-context learning exhibited by Transformers with the help of a simplified architecture and Markovian synthetic data. The authors show that experimentally the attention layers form statistical induction heads that help the model to implement an add-constant estimator based on the empirical counts of the input. They also show that a simplified transformer architecture can effectively represent such an estimator, and they provide a SGD convergence analysis for a minimal model.

**Strengths:**

The paper joins a line of works that aim at studying transformers with the help of synthetic data generated according to Markov distributions. Even if the model considered is significantly simplified, I believe that the theoretical and experimental insights provided by the paper are intriguing. The paper is also well written and the results seem correct to me, even if the notation can be improved, especially for the proofs in the appendix, that are not easy to follow.

**Weaknesses:**

The main limitations of the paper are: (1) the transformer architecture is heavily simplified (even if I believe that the results obtained in the paper should extend to more complex architectures); (2) the model used for the SGD analysis is, indeed, minimal. While limitation (1) is not so important to me, I am a bit concerned about limitation (2). I am not sure what is the actual utility of a SGD convergence analysis on such a minimal model. While I think that the representation result of Proposition 2.2 can be extended to a more complex model, I don't think that the SGD analysis of the minimal model would work for a more complex architecture, which would then need completely different and more sophisticated techniques.

**Questions:**

1. Do you think that an SGD analysis similar to the one carried out for the minimal model would carry over to a more complex architecture, closer to the actual transformer? If not, what are the main issues with working out such a analysis for, for example, the simplified transformer model of Equation (1)?
2. Your model architecture uses relative positional embeddings, as opposed to most state-of-the-art transformer models. Do you think that your analysis would be easily extended to a model with absolute positional embeddings?

---

> ### Author Rebuttal · Authors · 2024-08-07
>
> We thank the reviewer for their helpful review. Here we address the questions/concerns raised by the reviewer.
>
> 1. > Do you think that an SGD analysis similar to the one carried out for the minimal model would carry over to a more complex architecture, closer to the actual transformer? If not, what are the main issues with working out such a analysis for, for example, the simplified transformer model of Equation (1)?
>
> We believe that the analysis would mostly carry over to a more complicated architecture with one exception. Specifically, the softmax in the second layer attention proved intractable to analyze at the same time as the softmax in the first layer. Analyzing softmax is a common difficulty in transformer analyses, and why most theoretical results apply to linear attention only, we are happy to have managed to include the softmax in the first layer. Empirically, the training dynamics of the minimal model are very similar to that of the full model, the primary difference being that the phase transition isn't as sharp in the minimal model. With the softmax in the second layer added in, the phase transition becomes sharp like in the full model, but the tools do not currently exist to analyze this (specifically nested softmaxes on inputs not close to 0 or infinity).
>
> To create our minimal model, we started from a two layer attention-only disentangled transformer (using relative positional embeddings) and iteratively simplified parts that empirically did not affect the training dynamics. In this context, disentangled transformer (introduced in [Elhage et al](https://transformer-circuits.pub/2021/framework/index.html)) is like a normal transformer, except the output of each layer is appended after the residual. In our construction, the first layer only attends to positional embeddings, and the second layer ignores positional embeddings, so we set the first layer key matrix, $W_k^{(1)}$, to $0$, and removed positional embeddings from the second layer ($v^{(2)}=0$). Then we set the value matrices and query matrices to the identity in both layers. In the experiments, all optimizations such as layer norms and weight decay were removed, and the optimizer used was SGD. With all of these changes, the overall training dynamics were not changed much at all, depending on hyper parameters they could speed up training, but the same loss curve and phases were observed. To make analysis of gradient descent on the minimal model feasible, the softmax on the second layer had to be removed, which did make the phase transition less sharp.
>
> We believe an analysis that includes this softmax could confirm additional insights into why the phase transition is so sharp, and is a potential direction for future work.
>
> 2. > Your model architecture uses relative positional embeddings, as opposed to most state-of-the-art transformer models. Do you think that your analysis would be easily extended to a model with absolute positional embeddings?
>
> Empirically, the absolute positional embeddings add noise, but do not fundamentally change any of the results. The analysis might be a bit messier, but we would not expect it to be fundamentally different. Additionally, many state of the art models use non absolute positional embeddings, including the Llama series of models ([1](https://arxiv.org/abs/2302.13971), [2](https://arxiv.org/abs/2307.09288), and [3](https://arxiv.org/abs/2407.21783)).

---

> > ### Comment · Reviewer_oy83 · 2024-08-09
> >
> > Thank you for your comments. I will maintain my positive score.

---

### Comment · Area_Chair_9Bro · 2024-08-09
**Reviewers, please respond to the author rebuttals.**

Dear reviewers, this is your AC!

The authors have responded to your review. Could you please take a look at the rebuttal (and respond/edit your scores if need be)?
(Thank you if you have already done so)

Thanks!

---

### Decision · Program_Chairs · 2024-09-25

**Decision:**

Accept (poster)

**Comment:**

This paper studies the in-context learning capabilities of Transformers on an in-context learning task, where the model has to a learn a random n-gram model in-context. The paper studies two settings: a simplified Transformer architecture with only attention layers, and a "minimal" model that is more amenable to theoretical analysis.

 On the positive side, this paper studies an interesting synthetic setup (as studied by the concurrent works mentioned by the authors) that illuminates the inner workings of Transformers. On the negative side:
- the theoretical analysis is only performed on the minimal model, and while the authors argue that the behavioral similarity between the simplified Transformer and the minimal model justifies extending the analysis to the simplified model, this is a bit of a leap
- the synthetic experiments are extremely toy
- there are no real-world experiments (unlike some recent papers which use insights from synthetic tasks to improve actual language models).

Nonetheless, I think the contributions here are enough to justify acceptance.